# Seismic monitoring of small alpine rockfalls – validity, precision and limitations

Michael Dietze[1], Solmaz Mohadjer[2], Jens M. Turowski[1], Todd A. Ehlers[2], and Niels Hovius[1]

[1]GFZ German Research Centre for Geosciences, Section 5.1 Geomorphology, Potsdam, Germany
[2]University of Tübingen, Department of Geosciences, Tübingen, Germany

*Correspondence to:* Michael Dietze (mdietze@gfz-potsdam.de)

**Abstract.** Rockfall in deglaciated mountain valleys is perhaps the most important post-glacial geomorphic process for determining the rates and patterns of valley wall erosion. Furthermore, rockfall poses a significant hazard to inhabitants and motivates monitoring efforts in populated areas. Traditional rockfall detection methods, such as aerial photography and terrestrial laser scanning (TLS) data evaluation provide constraints on the location and released volume of rock, but have limitations due to significant time lags or integration times between surveys, and deliver limited information on rockfall triggering mechanisms and the dynamics of individual events. Environmental seismology, the study of seismic signals emitted by processes at the Earth's surface, provides a complementary solution to these shortcomings. However, this approach is predominantly limited by the strength of the signals emitted by a source and their transformation and attenuation towards receivers. To test the ability of seismic methods to identify and locate small rockfalls, and to characterise their dynamics, we surveyed a 2.16 km$^2$ large, near vertical cliff section of the Lauterbrunnen Valley in the Swiss Alps with a TLS device and six broadband seismometers. During 37 days in autumn 2014, ten TLS-detected rockfalls with volumes ranging from $0.053 \pm 0.004$ to $2.338 \pm 0.085$ m$^3$ were independently detected and located by the seismic approach, with a deviation of $81^{+59}_{-29}$ m (about 7 % of the average inter-station distance of the seismometer network). Further potential rockfalls were detected outside the TLS-surveyed cliff area. The onset of individual events can be determined within a few milliseconds, and their dynamics can be resolved into distinct phases, such as detachment, free fall, intermittent impact, fragmentation, arrival at the talus slope and subsequent slope activity. The small rockfall volumes in this area require significant supervision during data processing: 2175 initially picked potential events reduced to 511 potential events after applying automatic rejection criteria. The 511 events needed to be inspected manually to reveal 19 short earthquakes and 37 potential rockfalls, including the ten TLS-detected events. Rockfall volume does not show a relationship with released seismic energy or peak amplitude at this spatial scale due to the dominance of other, process-inherent factors, such as fall height, degree of fragmentation, and subsequent talus slope activity. The combination of TLS and environmental seismology provides, despite the significant amount of manual data processing, a detailed validation of seismic detection of small volume rockfalls, and revealed unprecedented temporal, spatial and geometric details about rockfalls in steep mountainous terrain.

# 1 Introduction

Rockfall is a dominant geomorphic process shaping the steepest slopes and landforms that constitute significant portions of mountainous terrain. Despite their small volumes ($10^{-1}$–$10^3$ m$^3$) in comparison with other mass wasting processes, such as rock avalanches ($10^2$–$10^5$ m$^3$) and rockslides ($> 10^6$ m$^3$) (Krautblatter et al., 2012), rockfalls can pose a significant hazard, due to their rapid evolution, high velocity and impact energy, and proximity to infrastructure. Thus, precise information on released volume, timing, location, dynamics and triggers is essential for understanding the underlying mechanisms, improving process based models, and to build robust mitigation and early warning systems. The unpredictable occurrence of rockfalls hinders detailed investigation of their dynamics and drivers under natural conditions. Direct observation of events is rare and restricted to, for example, the Yosemite Valley with thousands of camera-equipped tourists per day (Stock et al., 2013). Typical approaches to deliver information about rockfalls are deterministic and probabilistic susceptibility analysis, predictive modelling, a posteriori mapping of detachment zones, released volumes and pathways by aerial and satellite imagery or repeated terrestrial laser scan (TLS) surveying (Volkwein et al., 2011). The latter technique (Ring, 1963) provides high-resolution spatial data of topographic change attributable to rock detachment (e.g., Rabatel et al., 2008; Zimmer et al., 2012; Strunden et al., 2014), but is time consuming during recording and evaluation and primarily suited for monthly to annual lapse times. Over the integration time between two consecutive scans it is possible to identify spatial activity patterns, released volume ranges and magnitude-frequency relationships (Strunden et al., 2014). However, multiple rockfall releases from the same location cannot be resolved. Likewise, the relation between processes and external triggers remains obscured by the relatively coarse time resolution associated with many repeat TLS studies. Hence, insight into the individual stages of a single event (i.e., detachment, fall, impact and disintegration, duration, multiple failures) is not possible.

Seismic methods provide a solution for this shortcoming. Broadband seismometer networks have been used to detect and locate a wide variety of Earth surface processes, such as landslides (e.g., Dammeier et al., 2011; Burtin et al., 2013; Ekström and Stark, 2013), rockslides and rock avalanches (e.g., Hibert et al., 2011; Lacroix and Helmstetter, 2011), debris flows (Burtin et al., 2014) and bed load transport in rivers (e.g., Burtin et al., 2008; Gimbert et al., 2014). This emerging research field as well as using seismic noise cross-correlation methods to investigate the states and changes of subsurface conditions are referred to as environmental seismology (Larose et al., 2015). Current studies (e.g., Helmstetter and Garambois, 2010; Hibert et al., 2011; Burtin et al., 2014; Farin et al., 2015) have focused on monitoring activity at catchment or sub-catchment scale, usually either with limited validation against independent data, focusing on detachment volumes above $10^3$ m$^3$, or working under very controlled, laboratory-like experimental conditions.

Combining TLS and seismic data may provide essential and complementary information on rockfall dynamics and characteristics. This could allow assessing the performance of the seismic approach in terms of correctly identified events, missed events, additional events and spurious events. Further, the combined approach could contribute information beyond the TLS data, such as the existence of rockfalls from the same location but subsequent activity periods or insight into individual stages of a rockfall sequence. In this study, we investigate the validity of environmental seismology to detect and locate rockfall events

that are independently identified by TLS surveys in a steep valley of the European Alps. This validation includes exploring the limits of seismic detection in terms of rockfall size and the accuracy of individual event location.

## 2   Study area

The Lauterbrunnen Valley in the central Swiss Alps is a deglaciated U-shaped valley. It is flanked by up to 1000 m high, Mesozoic limestone cliffs with sometimes almost vertical walls (88.5 °) and several hanging valleys that host more than 70 waterfalls. Talus slopes at the base of the cliff, reaching around 150 m above the valley floor, argue for substantial and sustained rockfall. The steepest wall section separates the town of Mürren above the cliff from the town of Lauterbrunnen in the valley (Fig. 1). Our study focused on this wall, which has minimal snow and vegetation cover throughout the year. The surrounding area contains further rockfall-prone locations that can deliver rockfall signals, such as the steep slopes of the Chänelegg and the ridge south of the Ägertenbach (Fig. 1 a). The steep topography of the Lauterbrunnen Valley with a few small ledges (Fig. 4 b) implies a significant free fall phase of detached rocks, followed by rockmass impacts on the cliff face or the talus slopes below, eventually grading into moderate translocation processes on the less than 250 m long depositional areas. Rockfall activity in the Lauterbrunnen Valley has been monitored by repeated TLS since 2012 (Strunden et al., 2014), yielding 122 detected rockfalls (523.72 $m^3$ in total) over an 18 month investigation period. These events appear to be evenly distributed throughout valley walls (15.13 events per year and $km^2$) with most frequent events being smaller than 1 $m^3$.

## 3   The seismic view on rockfall

The seismic approach to studying Earth surface processes (Fig. 2, i.e., event 7 from Table 1) utilises the ground motion recorded by a network of sensors. These signals can be studied in the time domain (i.e., time series of ground velocity) and frequency domain (i.e., the frequency spectrum of the entire signal), or in combination (i.e., spectrograms, stacked spectra of time slices of the signal). A rockfall event manifests as a series of short and long pulses of ground velocity above the ambient background noise level (Fig. 2 a), with characteristic frequency contents over the entire frequency band above 5 Hz (Fig. 2 b), usually dominated by the 10–30 Hz band (e.g., Hibert et al., 2014). This characteristic pattern makes rockfalls distinct from other seismic sources, such as earthquakes and anthropogenic noise. The individual pulses and their spectral properties can be interpreted genetically, e.g., as successive rock mass impacts, fragmentation and subsequent slope activity (e.g., Burtin et al., 2014; Hibert et al., 2014). Each signal pulse, emitted at a source location, travels predominantly as a surface wave (e.g., Dammeier et al., 2011; Levy et al., 2015; Burtin et al., 2016) with a finite velocity. Thus, in a homogeneous medium, the seismic signal arrives at different seismic stations at different times and with systematic, frequency- and distance-dependent changes of the signal properties. These property changes can be significantly altered due to heterogeneous rock and structure characteristics in natural environments. Nevertheless, the time offsets with which signals are recorded at the stations allows finding a location in space that best explains the overall spread of signal arrival times at all stations. Thus, seismic signals have

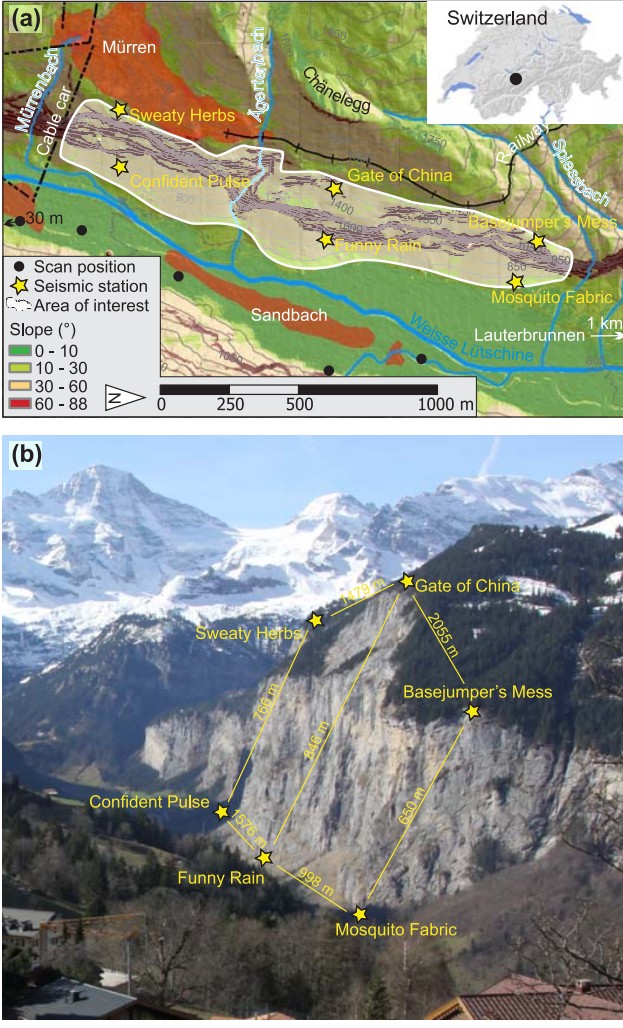

**Figure 1.** The study area Lauterbrunnen Valley. (a) Schematic map with location of seismic stations, TLS positions and anthropogenic noise sources (settlements, technical infrastructure). (b) Photograph of the instrumented east-facing cliff face of the Lauterbrunen Valley with the Breithorn and Tschingelhorn in the background. Seismic stations (yellow stars) are separated by 1200 m on average.

the potential to deliver unique, important information about rockfall dynamics and location, if comparison with independent data can confirm the validity of the approach.

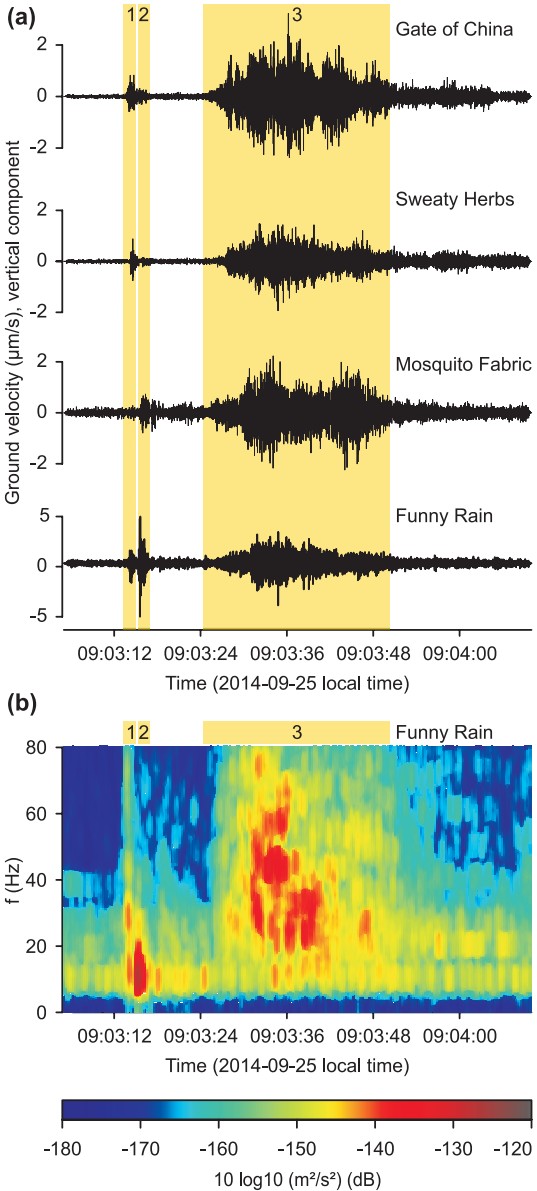

**Figure 2.** Anatomy of a $0.891 \pm 0.038$ m$^3$ large rockfall event (event 7 from Table 2). (a) Seismic waveforms (filtered between 1 and 90 Hz) of four stations (see Fig. 1 for locations). (b) Power spectral density estimate of station "Funny Rain". Two distinct, short seismic activity phases (yellow polygons 1 and 2) are followed by an emergent and prolonged period of activity (yellow polygon 3) after 7.5 s of calm.

## 4  Methods

### 4.1  Equipment and deployment

High resolution point clouds with a limit of detection (i.e., the smallest resolvable length fraction at the cliff surface) of about 11 cm were generated by TLS, using an Optech ILRIS-LR terrestrial light detection and ranging scanner with a scan frequency
of 10 kHz and a reflectivity of 80 % at 3 km distance. Scans were recorded during two field campaigns on 22 September 2014 and 28 October 2014. The TLS data collection and processing approach used in this study is identical to that of previous work conducted in the same study area (for details see Strunden et al., 2014). To ensure sufficient overlap and to avoid topographic shading effects, the study area was scanned from five different positions (see Fig. 1 a). Seismic activity was measured by six Nanometrics Trillium Compact 120s three component broadband seismometers. The ground velocity signals were recorded
with Omnirecs Cube ext[3] data loggers, sampling at 200 Hz, with gain set to 1 and a GPS flush time 30 minutes. Deployment sites were chosen to optimise the potential for rockfall location along the east-facing rock wall below the town of Mürren. Stations were separated from each other laterally by 1000–2050 m and vertically by 650–850 m. Three stations were deployed along the upper limits of the talus slopes at the cliff base and three stations on top of the cliff (Fig. 1). Each seismic sensor was installed in a small hand dug pit at 30–40 cm depth. Seismic activity was recorded for 89 days, between 1 August and
28 October 2014. In this study only the period bracketed by the two TLS surveys is analysed (22 September–28 October). For event location a digital elevation model (DEM) of the wider study area with 5 m grid size (swissALTI3D) was used, transformed to the UTM coordinate system and resampled to 10 m grid size.

### 4.2  TLS data processing

Point clouds were processed with the "Joint Research Center 3-D Reconstructor 2" software (Gexcel, 2017), adjusted manually
and merged using visualy unaffected control points along the cliff and a best fit algorithm to minimize differences in the overlapping data. Rockfall detachment locations and volumes were calculated from the two data sets using the inspection tool and the cut and fill algorithm. Photographs recorded during scanning were used to confirm that the detected volume changes were not caused by processes other than rockfall (e.g., vegetation growth). Measurement uncertainty was estimated based on scan differences from stable control regions (for details see Strunden et al., 2014). Detachment area coordinates were obtained
by georeferencing the rasterised point cloud data on referenced topographic maps and orthoimages. Given the typical rockfall volumes < 1 m$^3$ (Strunden et al., 2014), location uncertainty should mainly result from the georeferencing process and is quantified by the root mean square error (RMSE). All location coordinates were rounded to the full meter and transformed to the UTM coordinate system.

### 4.3  Seismic data processing: Event detection

A single seismic station records 200 samples per second and geometric signal component, resulting in more than 311 million measured values per day. Hence, potential rockfall events must be identified (picked) automatically from the stream of data

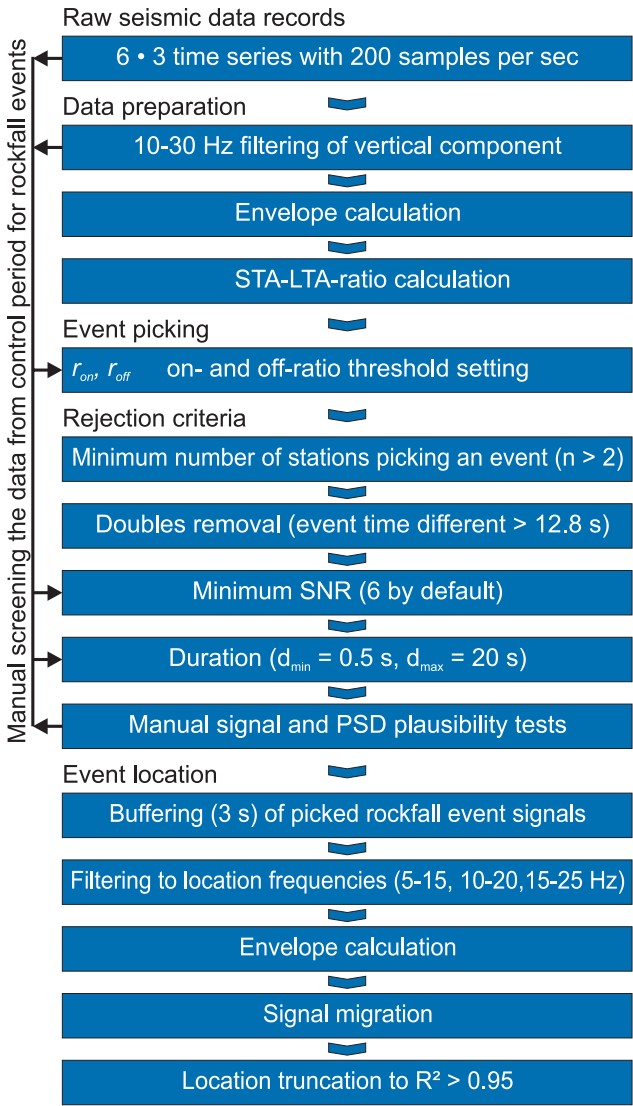

**Figure 3.** Schematic flow chart of the work flow for seismic data analysis. Arrows along left side of the boxes indicate utilisation of control period data.

before they can be located and described (Fig. 3). However, for rockfall events with volumes usually below 1 m$^3$ (Strunden et al., 2014) it is challenging to find reasonable parameter settings for any picking algorithm. Therefore, the seismic time series of all operating stations were manually screened during a control period, 22 September–1 October, to find reference events for parameter definitions.

5     We used an STA-LTA-ratio algorithm (Allen, 1982), calculating the continuous ratio between a long term average (LTA) and a short-term average (STA) of the signal envelope. When the onset of an event is recorded, it will not affect the LTA value but

have a significant effect on the STA value, thus increasing the ratio. When the seismic signal returns to background, the STA values approach the LTA value again, which lowers the ratio towards one. The STA-LTA-ratio picker thus has four relevant parameters: the lengths of the STA window and LTA window, a threshold value to define the start of an event and another threshold value to define the end of an event (Fig. 3).

In the case of the Lauterbrunnen Valley, the STA window was set to 0.5 s and the LTA window to 180 s, based on the experiences of Burtin et al. (2014) from another steep mountainous catchment. The window lengths obviously affect the number and timing of the initially picked events. Thus, to be sensitive to short-lasting and low-magnitude rockfall events, we used this short STA versus long LTA value. The threshold values for defining the start and end of an event were adjusted based on manually identified events from the control period (section 5.2). The LTA value was set to constant after an event onset to

avoid spurious changes of the ratio for long lasting events (Burtin et al., 2014).

     The STA-LTA-ratio algorithm was applied to the bandpass-filtered (third order Butterworth filter) envelope of the vertical component signal of the central cliff top station "Gate of China" (Fig. 1, 3). The filter cut-off frequencies were set to 10 and 30 Hz to isolate the typical frequencies of rockfalls and rock avalanches (Helmstetter and Garambois, 2010; Hibert et al., 2014; Burtin et al., 2014; Zimmer and Sitar, 2015). Since a significant rockfall should be detected by more than one station,

we require that all events interpreted are identifiable at this central station. Furthermore, this station was chosen because of its remote location, away from potential sources of anthropogenic and fluvial noise, in order to reduce the initial number of spurious detections. Events that were not co-detected by at least two other stations within a time window of 1.75 s were removed from the data set (Fig. 3). The value of 1.75 s corresponds to the maximum travel time of a seismic S-wave within the entire seismic network when using a low S-wave velocity in limestone of 2000 $ms^{-1}$ (Bourbie et al., 1987). This value is

also similar to the apparent velocities of local earthquakes and rockfalls as discussed by Burtin et al. (2009) and Helmstetter and Garambois (2010). In general, a rockfall event can consist of multiple block releases and impacts, and subsequent hillslope activity, all at different locations. Not accounting for such effects by setting the 1.75 s criteria would introduce artifacts that bias the subsequent location approach. Similarly, if two consecutive picked events showed a time offset smaller than 12.8 s, then only the first one was kept. The selected value of 12.8 s corresponds to the maximum possible free fall time of a rock mass

from the top of the highest cliff part. This ensured that rockfalls with multiple impacts were not identified as separate events. However, this also implies that in the case of two unrelated rockfalls, occurring within this time window, the latter one would be ignored.

     Further options to reduce false detections can be setting thresholds for minimum and maximum event duration, signal amplitude variance throughout the network, comparison with existing catalogues (e.g., the Swiss earthquake catalogue), and

signal-to-noise (SNR) ratio (Burtin et al., 2014, 2016), the latter being the ratio of the maximum and average value of the signal envelope of an event. However, all these thresholds must be adjusted to an existing data set of potential rockfall events and their effects should be inspected. For the subsequent analysis, minimum and maximum duration as well as SNR ratio were used as rejection criteria, with parameters adjusted based on the control period (Fig. 3, section 5.2).

     The waveforms of all remaining events were inspected manually for plausibility, validity and the possibility to locate their

source. This included the following criteria (see Fig. 2 for an example of how the criteria are matched): i) they should not

exhibit the typical features of earthquakes, such as distinct P- and S-wave arrivals, a long coda (i.e., the exponentially decaying tail of the signal), frequencies below 2 Hz, and similar amplitudes at all seismic stations for low frequencies, ii) they must show significant differences in signal amplitudes due to the source receiver distance-related attenuation within the network iii) they should either exhibit the presence of one or more erratic peaks in the seismogram as the result of impulsive impacts (Zimmer and Sitar, 2015) or show an avalanche-like emergent signal, i.e., several seconds rise time of the signal from background, followed by a long decay into background noise after reaching a maximum amplitude (e.g., Suriñach et al., 2005; Vilajosana et al., 2008; Zimmer et al., 2012). The temporal evolution of potential event signals was further inspected using power spectral density (PSD) estimates. These were calculated according to the method of Welch (1967) with moving time windows of 1.4 and 1.1 s to generate the spectra, each with an overlap of 90 %, and the individual spectra were corrected using the multitaper method. Rockfall events typically exhibit a burst of seismic energy over a wide frequency range during the first impulsive impact, possibly followed by subsequent activity in the 10–30 Hz frequency band (Vilajosana et al., 2008; Dammeier et al., 2011; Hibert et al., 2011). The detected potential events should agree with these observations. All successfully evaluated events were used for subsequent analyses.

### 4.4 Seismic data processing: Event location

Locating the source of the seismic signals emitted by rockfalls can be challenging due to the emergent onset of events, superposition of many impact signals, significant high-frequency content, missing constraints on specific seismic wave types and differences between waveform properties at different stations. The latter is due to the preferential signal attenuation of higher frequency waves, fragmentation of rocks during impact and changing amplitudes with time due to the moving source approaching or passing by a station (Burtin et al., 2013, 2016). Approaches that use the full waveform (Lacroix and Helmstetter, 2011) or its envelope (Burtin et al., 2013) are more appropriate to locate the source of seismic signals resulting from such processes. They are based on calculating average cross-correlations of signal pairs, each shifted by the time delay experienced due to the distance of a grid cell to a seismic station. The grid pixel with the highest overall correlation value is deemed to be the most likely source location. When encountering moving sources, signal migration needs to be performed for each impact signal separately to avoid "blurring" of the location estimate. The probabilistic signal migration approach further requires constraints on the average seismic wave velocity within the area of interest, a suitable frequency window for processing the signals and a topographic correction of the ray paths (Burtin et al., 2013).

Velocity tests were performed with two approaches. For all 37 picked potential rockfall events, the seismic wave velocity within rock was changed between 700 and 4000 $ms^{-1}$ to inspect its influence on the average cross correlation strength of the signal envelopes at different stations. In a further independent approach we used the TLS-based rockfall detachment locations to evaluate the effect of the different wave velocities considered, based on the average difference between the seismic and TLS locations. This second approach is only possible when independent information of rockfall locations are present and can also be seen as a validation of the first approach.

Similar to the velocity, the frequency band used in the location routine can have an influence on the location estimate. Both parameters are interconnected and may be optimised with respect to the overall highest cross-correlation value of the

location estimate. However, in this study the average seismic wave velocity is regarded a global, spatially and temporally constant parameter and was not adjusted for different frequency bands. For rock avalanches along the steeply inclined slopes of the Illgraben catchment and a widely distributed network of nine seismometers, Burtin et al. (2013) chose the frequency window with the highest signal-to-noise ratio. In the case of the Lauterbrunnen Valley, the seismic signals were much more
heterogeneous among the stations. There was no common frequency with high signal-to-noise ratio at all stations. Hence, we used fixed windows of 5–15, 10–20 and 15–25 Hz, depending on the dominant frequency range of the first impact signals. Usually, an event could be located at comparable positions with all three frequency windows. In that case, the window with the highest cross-correlation value was chosen. In cases where none of the three windows resulted in a stable location along the cliff face or other potential rock release zones inside the study area, the frequency windows were adjusted manually based on
the dominant frequency range in the PSD. In a second step, the frequency windows of all events were subsequently adjusted manually to minimise the difference between the seismic and TLS-based location estimates of rockfall events. Obviously, this optimisation is only possible when independent location constraints are present and will have different frequency values for each event. Thus, it is used here to evaluate the appropriateness of the fixed frequency window approach and to explore the maximum possible location precision available with the data, methodology and landscape setting of this specific experiment.

Topography correction is necessary because rockfalls and other gravitational mass wasting processes generate surface waves that propagate following the topography (Dammeier et al., 2011; Hibert et al., 2014; Lin et al., 2015; Burtin et al., 2016). The results of this correction were stored in distance maps. These are station-specific grids of the same resolution as the input DEM (10 m) where the cumulative direct distance of each pixel to a seismic station has been modified by that part where the direct distance was above the actual surface elevation (Burtin et al., 2014). Specifically, the distance between each pixel and station
is approximated as a straight line of pixel-sized segments in three dimensional space (xyz vectors) and whenever the z value (elevation) of a segment is above the DEM-based z value, it is replaced by the latter. The final distance is calculated as the sum of vector magnitudes. To ensure that topographic modification of the wave path is resolved, it is important that the wavelength (i.e., the ratio of wave velocity and frequency) is several times smaller than the average distance between seismic source and the recording station. For typical wave velocities in limestone between 2000 and 3300 ms$^{-1}$ (Bourbie et al., 1987; Helmstetter
and Garambois, 2010) and useful frequencies of 10–30 Hz, the wavelengths are a few hundred metres, which is adequate for the average distance between seismic stations (Fig. 1 b).

All picked events were clipped with a buffer of 3 s before and after the event and then migrated. Locations with a cross-correlation value $R^2$ below the 0.95 quantile were removed and the remaining values were normalised between 0 and 1. Events located along the margin pixels of the distance map of the study area were rejected. Only events inside the area of interest
(Fig. 1) were used for validation. The threshold quantile value of 0.95 to clip location areas is arbitrary though in the range of values from the literature (Burtin et al., 2014). The effect of this value on the number of rockfall locations inside the resulting uncertainty polygon was tested by changing the value from 0.9–1.0 and recording the number of TLS-based detachment locations and corresponding downslope trajectories, which remained inside the uncertainty polygons.

Location differences $\Delta P_{max}$ were calculated as the minimum planform Euclidean distance between the highest value of
the seismic location estimate ($P_{max}$) and the downslope trajectory line of the corresponding TLS-based detachment pixel.

The direction of the trajectory line was defined by the average cliff face azimuth (99 $\pm$ 44°). This approach was chosen because seismic signals can only be emitted at the detachment zone or rockfall impact sites below it, and since the cliff face is nearly 90° steep there is a high likelihood that the rock mass will follow the line of steepest descend without much deviation. Uncertainties arising from deviations of the rock mass from this line could not be accounted for.

All seismic analyses were performed in the R environment for statistical computing (R Development Core Team, 2015) (version 3.3.1) using the packages eseis (Dietze, 2016), sp (Pebesma and Bivand, 2005; Bivand et al., 2013; Pebesma and Bivand, 2016) and raster (Hijmans, 2016).

## 5  Results

### 5.1  Lidar-detected rockfalls

Between 22 September and 28 October, ten rockfall events were detected by TLS. The events were spread over the entire monitored part of the cliff, but the southern section, near stations "Sweaty Herbs" and "Confident Pulse", hosted 50 % of all events. The smallest detected rockfall (event 5 in Table 1) had a volume of $0.053 \pm 0.004$ m$^3$ while the largest rockfall (event 10 in Table 1) had a volume of $2.338 \pm 0.085$ m$^3$. The average volume of rockfalls in this period was 0.482 m$^3$. A summary of all rockfall events including location coordinates based on TLS and seismic data is shown in Table 1. With only one exception (event 6), all rockfalls detached from the lower part of the cliff, some almost at the base (Fig. 4 b, Table 1). The georeferenced RMSE in the event locations was between 4.8 and 17.5 m. The range in RMSE values calculated depends on the number of identified ground control points (between 8 and 17 per scene) as well as the size and perspective of the referenced image.

### 5.2  Continuous seismic data processing

Over the entire monitoring period there were always at least four seismic stations operating simultaneously. Due to topographic shielding, the basal stations needed several days after deployment and maintenance to receive a GPS signal, necessary for time synchronisation. Two seismic stations failed during the monitoring period ("Basejumpers Mess" on August 29 and "Confident Pulse" on September 27), due to progressive sensor tilting caused by slope movement or sediment settling. However, the remaining stations provided sufficient data for detection and location of events, i.e., all event descriptions are based on data from four seismic stations.

Manual screening of seismic records during the control period (22 September and 1 October) yielded evidence of two rockfalls, events 7 and 10 of the final data set (Table 2). One of these rockfalls (event 7, Fig. 5 b) generated two short, distinct bursts of seismic energy, less than 2 s apart, followed by a rise of the seismic signal about 7.5 s later (see Fig. 2 for details). The first burst contains frequencies between 30 and 60 Hz, while the second peak mainly has frequencies below 20 Hz. The subsequent strengthening signal is again dominated by frequencies between 30 and 80 Hz. The entire sequence was recorded by all operating stations, though with different amplitudes, from about $\pm$ 0.38 $\mu$ms$^{-1}$ at station "Sweaty Herbs" to $\pm$ 4.9

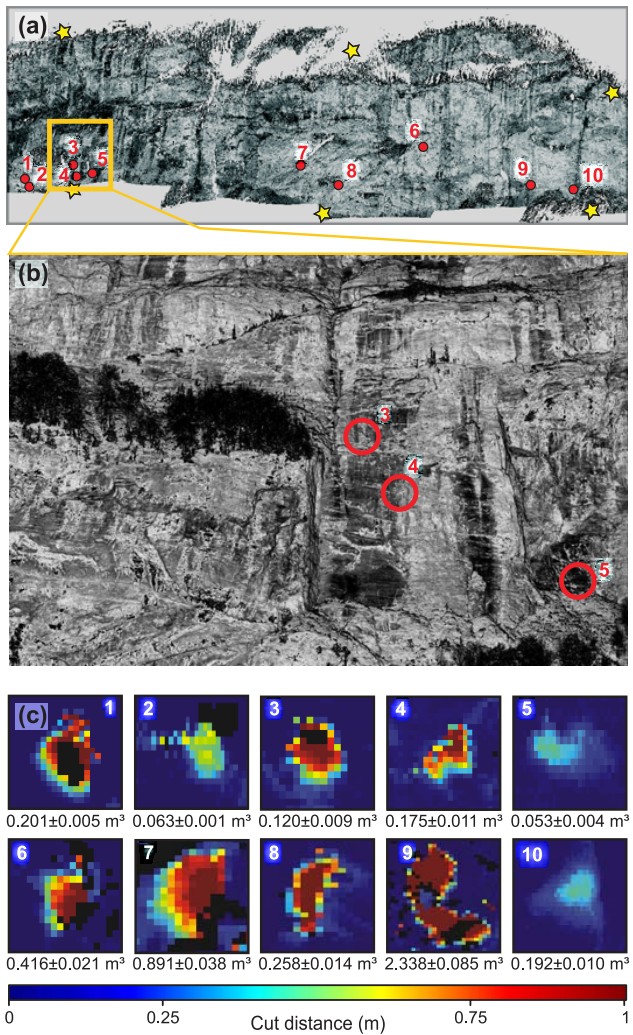

**Figure 4.** Rockfall detachment zones determined from TLS mapping. (a) Overview (aligned point cloud data) of the about 2.7 km long, instrumented east-facing stretch of the Lauterbrunnen Valley with rockfall detachment zones (red dots) and seismic stations (yellow stars, station names and distances see Fig. 1). (b) Close-up of the southern rock wall section with the detachment zones of events 3 – 5 at elevations less than 100 m above the talus slope. (c) Boxes show rockfall detachment patterns on the rock wall. Released rock volumes and uncertainties are given below each box. Event numbers are the same as in (a) and Tables 1 and 2)

.

$\mu$ms$^{-1}$ at station "Funny Rain". The maximum time offset between event onsets at the stations was 0.51 s. The STA-LTA-ratio values reached up to 7 for the first two peaks and decreased below 2 before grading to the next rise.

Based on the above characteristics of event 7 and similar properties for event 10 from the control period, the parameters for event picking of the entire data set were defined, i.e., the STA-LTA-ratio threshold to define the start of an event was set to 5,

**Table 1.** Rockfall location summary. Subscript TLS denotes UTM coordinates from aligned TLS point cloud data. Subscript seis denotes coordinates based on seismic signal processing, i.e., site/point of the highest location probability ($P_{max}$). Ranges of z-coordinates are determined as min-max range of a 3 by 3 pixel matrix around the detected location. P diameter is the greatest lateral diameter of the location uncertainty polygon (Fig. 9). $\Delta P_{max}$ is the deviation of the most likely seismic location estimate from the rockfall trajectory as determined from TLS surveys. The values outside parentheses give deviations with default settings, values in parentheses give smallest possible deviations with optimised location frequency windows (only possible when independent location data is available).

| ID | $x_{TLS}$ (m) | $y_{TLS}$ (m) | $z_{TLS}$ (m) | $V_{TLS}$ (m$^3$) | $x_{seis}$ (m) | $y_{seis}$ (m) | $z_{seis}$ (m) | P diameter (m) | $\Delta P_{max}$ (m) |
|---|---|---|---|---|---|---|---|---|---|
| 1 | 415511 | 5156535 | 964–1036 | $0.201 \pm 0.005$ | 415485 | 5156551 | 1063–1119 | 860 | 760 (31) |
| 2 | 415523 | 5156542 | 952–1022 | $0.063 \pm 0.006$ | 415505 | 5156541 | 1005–1063 | 792 | 50 (18) |
| 3 | 415541 | 5156844 | 1084–1138 | $0.201 \pm 0.005$ | 415515 | 5156841 | 1141–1192 | 943 | 27 (27) |
| 4 | 415566 | 5156845 | 1018–1100 | $0.175 \pm 0.011$ | 415505 | 5156871 | 1184–1218 | 968 | 92 (66) |
| 5 | 415591 | 5156934 | 1009–1062 | $0.053 \pm 0.004$ | 415635 | 5156991 | 999–1054 | 587 | 147 (63) |
| 6 | 415950 | 5158213 | 1170–1314 | $0.416 \pm 0.021$ | 415965 | 5158241 | 1182–1224 | 687 | 21 (21) |
| 7 | 415952 | 5157829 | 1048–1123 | $0.891 \pm 0.038$ | 416015 | 5157781 | 907–927 | 858 | 117 (37) |
| 8 | 416005 | 5157897 | 916–1026 | $0.258 \pm 0.014$ | 416015 | 5157891 | 889–954 | 614 | 251 (4) |
| 9 | 416116 | 5158797 | 919–1002 | $0.192 \pm 0.010$ | 416065 | 5158811 | 1117–1217 | 498 | 70 (53) |
| 10 | 416037 | 5158649 | 979–1114 | $2.338 \pm 0.085$ | 416095 | 5158691 | 922–939 | 361 | 60 (52) |

the threshold for defining the end of an event to 3. Note that this approach does not yield a correct start and end time. However, the location approach is not based on exact onset times but is used with the addition of a 3 s wide buffer before and after an event. The minimum SNR of an event at the picking station "Gate of China" was set to 6.

The instrumented study area comprises many further environmental sources that generate seismic signals with frequencies above 1 Hz. Fig. 5 a shows a 24 h PSD as an example. From 4 am to 9 pm (UTC time, i.e., -2 h to local time) there are pulses of seismic activity in the 5–80 Hz range, occurring every 20 minutes. Until 2 am there is continuous activity with frequencies above 30 Hz and over the day there is a progressively decreasing signal between 5 and 15 Hz, which in general depicts the runoff of the Weisse Lütschine (FOEN, 2017), the main river draining the Lauterbrunnen Valley, and is in agreement with the seismic signature of turbulent water flow (Gimbert et al., 2014). Around 2:45 am, 5:10 am and 5:50 pm and 6:05 pm there are seismic events with very low frequency content (maximum energy below 2 Hz). Fig. 5 c shows that the seismic properties of all these other sources can be very similar to the waveforms of rockfalls. Between 4 am and 9 pm (UTC time) a train runs every 20 minutes between Mürren and the cable car station of Lauterbrunnen. The passage of this train is recorded in a repeating succession of spikes of seismic energy in the PSD from Fig. 5 a. Although this signature is easily discernible because it repeats at expected times during the day (i.e., Swiss trains always run on time), it also shows two distinct peaks that cross the STA-LTA-ratio start and end thresholds for rockfall detection, and it shows similar amplitudes and amplitude differences between the recording stations. Also the SNR values are comparable with those of rockfalls. The second panel of Fig. 5 c shows the impact of rain drops on the ground above the seismic sensor. Attribution of this signal to rain drops is based on the notion that these

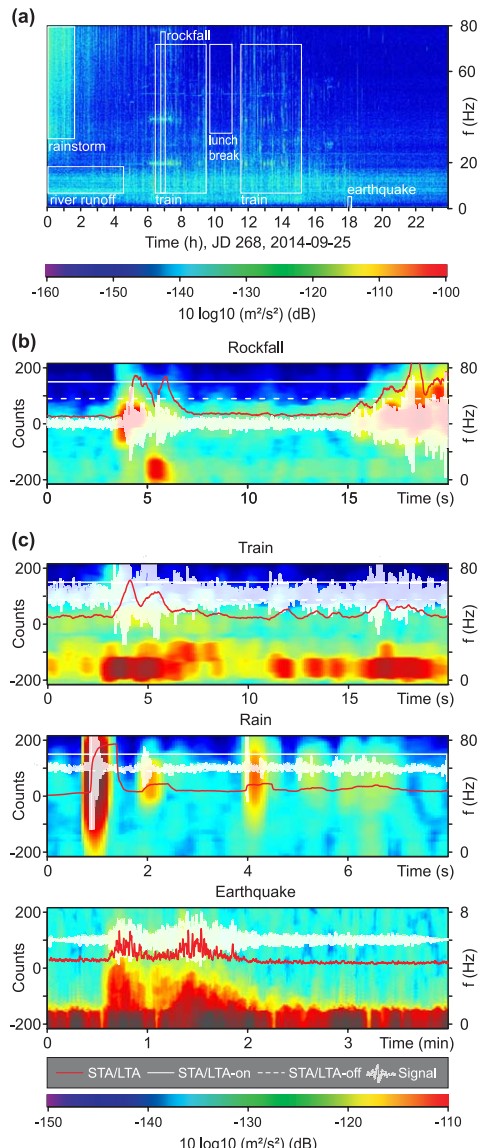

**Figure 5.** Example day (25 September 2014) showing seismic characteristics of environmental sources in the Lauterbrunnen Valley. (a) 24 h PSD with interpreted sources indicated. Data recorded at station "Mosquito Fabric" and filtered between 1 and 90 Hz. (b) Seismic record of rockfall event 7 (Table 2). (c) Seismic records of other sources registered by the station "Gate of China". Note change in axes scales for the earthquake event. (b) and (c) contain the PSD (background image, colour bar applies to (b) and (c)) and waveform data (semitransparent line graph) as well as the picker algorithm characteristics (STA-LTA-ratio, "on"- and "off"-thresholds).

irregular short pulses only occurred during rainy conditions (Fig. 6 a and b) and, furthermore, were predominantly registered by stations under forest cover in contrast to sensors deployed at grass covered sites (Fig. 6 c versus d). We attribute this

phenomenon to trees collecting small rain drops and releasing them after some time as larger drops. Trees continue to release such drops even after the atmospheric rain input has stopped. In contrast, grass covered areas receive the precipitation directly and are subject to systematically smaller drops, especially during gentle rain events. The irregular occurrence of the seismic pulses make an origin due to passing animals or humans unlikely, as one would expect a growing and decreasing amplitude during approaching, passing and leaving the station (a signature inherent to many base jumpers hiking past the stations on top of the cliff during sunny days). The signal of a raindrop is also similar to the rockfall signal although it contains seismic energy over nearly the entire frequency range and lasts less than half a second. Such signals can trigger the STA-LTA-ratio algorithm if they were recorded by chance at more than two stations within the defined maximum time window of 1.75 s. The last panel of Fig. 5 c shows an earthquake. The signal of this tele-seismic event is dominated by frequencies below 4 Hz and lasts more than one minute. There are also local earthquakes in the seismic records that show a more sudden onset, contain higher frequencies and last much less than a minute. But all earthquake signals are clearly different from rockfalls. Their waveforms usually show the distinct arrivals of P- and S-waves and a coda, their PSDs exhibit a significant portion of energy below 10 Hz, and their waveforms and spectral properties are relatively uniform among records of the different seismic stations.

**Table 2.** Rockfall events detected by seismic monitoring. IDs correspond to those in Table 1. Duration as estimated from signal wave form interpretation (not including subsequent talus slope activity). SNR denotes range of signal-to-noise ratios among all recording stations. $f_{default}$ describes the default frequency range for location, $f_{opt}$ denotes the frequency range after optimisation. $A$ is the amplitude range among the stations.

| ID | Time (UTC) | duration (s) | SNR | $f_{default}$ (Hz) | $f_{opt}$ (Hz) | $A$ ($nms^{-1}$) |
|----|------------|--------------|-----|--------------------|-----------------|-------------------|
| 1 | 2014-10-12 22:45:50 | 1 | 8.7–25.9 | 10–20 | 10.0–23.0 | 1356–11945 |
| 2 | 2014-10-15 01:58:32 | 4 | 5.2–49.4 | 10–20 | 11.0–21.0 | 1062–4128 |
| 3 | 2014-10-20 19:11:09 | 5 | 10.7–35.8 | 10–20 | 10.0–20.0 | 619–2405 |
| 4 | 2014-10-20 15:05:34 | 7 | 14.0–55.86 | 15–25 | 16.0–26.0 | 722–3229 |
| 5 | 2014-10-22 11:47:28 | 2 | 5.7–11.9 | 10–20 | 11.0–19.9 | 1442–3831 |
| 6 | 2014-10-02 17:59:50 | 4 | 6.48–11.76 | 5–15 | 5.0–16.0 | 1055–2077 |
| 7 | 2014-09-25 07:03:13 | 6 | 7.5–19.9 | 10–20 | 2.8–5.6 | 962–5980 |
| 8 | 2014-10-26 20:08:45 | 2 | 6.0–14.2 | 10–20 | 7.0–13.0 | 1277–306905 |
| 9 | 2014-10-17 00:09:25 | 8 | 5.5–11.1 | 5–15 | 4.7–15.2 | 828–1806 |
| 10 | 2014-10-01 09:23:05 | 10 | 17.0–59.1 | 5–35 | 1.0–35.0 | 3123–4491 |

Thus, to eliminate false events picked by the STA-LTA-ratio approach the minimum duration of an event was set to 0.5 s to remove rain-related picks and the maximum duration was set to 20 s to remove earthquakes. The minimum average SNR value among all stations was set to 6. The STA-LTA-ratio approach yielded a total of 2175 potential events. After application

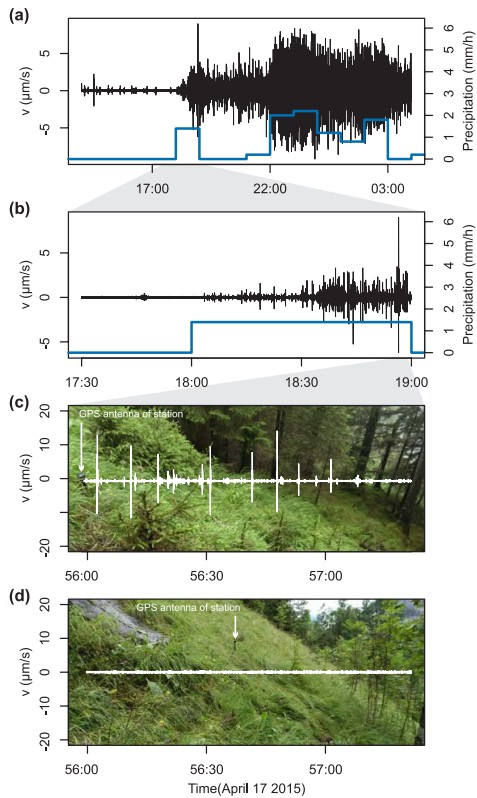

**Figure 6.** Seismic signal characteristics during a gentle rain event without windy conditions (hourly meteorological data from Meteomedia station in Mürren). Panels a to c show the vertical component signal (filtered between 1–90 Hz) of station "Basejumpers Mess". Panel d shows the same time interval as c but for station "Funny Rain". Background images of c and d show the deployment situation of the two stations under a dense coniferous forest cover and on grass land, respectively. Note overall increase in seismic signal amplitudes during the rain event and and short irregular signal pulses only under forest cover, interpreted as impacts of large drops collected and amalgamated by the trees. Trees continue to release drops even after the precipitation record shows no further atmospheric rain input (a).

of the automated rejection criteria the number decreased to 511. These 511 events had to be manually screened and included 455 spurious or unknown events, 19 short earthquakes and 37 potential rockfall signals. The most common spurious event type was associated with train traffic. This type of signal could not be eliminated by any automatic routine and had to be removed manually. The remaining earthquakes had an average STA-LTA-based duration of $11.9^{+4.6}_{-4.0}$ s (median and quartiles) and were also removed manually. The 37 detected potential rockfall events had STA-LTA-based durations of $4.7^{+2.8}_{-2.0}$ s. Several of the potential rockfall events had very weak seismic signals, with average SNRs below 8 (eight cases) but the majority generated average SNRs of $11.2^{+2.8}_{-2.6}$.

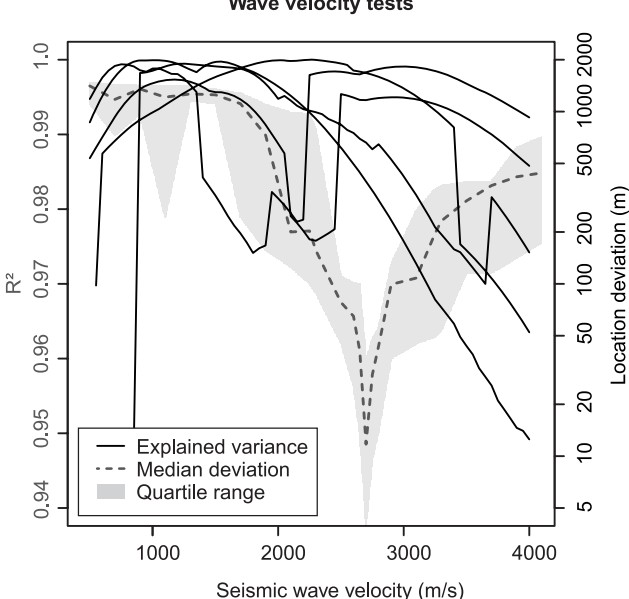

**Figure 7.** Tests of the most likely average value for the seismic wave velocity. Black solid lines show location approach correlation coefficient (average of all $R^2$) for velocity values ranging between 700–4000 $ms^{-1}$ for all events that reached an $R^2 > 0.94$. The dashed grey line (median) and shaded area (interquartile range) depict deviation of seismically detected from TLS-based event locations. Both measures point at 2700 $ms^{-1}$ as the most likely average seismic wave velocity in the study area. The secondary $R^2$ maximum at lower velocities did not yield locations inside the area of interest despite high $R^2$ values.

## 5.3 Seismic wave velocity estimate

A necessary step for successful location of the potential rockfall events was to find a plausible estimate of the average seismic wave velocity (Fig. 7). Both approaches, optimising the average location estimate value (i.e., $R^2$ at $P_{max}$) and minimising the difference between seismic location and TLS-based coordinates, point at a common value around 2700 $ms^{-1}$. While for the latter approach the velocity range with minimum offsets is narrow, with not much argument for an uncertainty range, there is no such clear result for the former approach. The solid black lines in Fig. 7 show two velocity ranges with high $P_{max}$ values, between 1000 and 1800 $ms^{-1}$ and between 2200 and 3000 $ms^{-1}$. Due to the recent deglaciation and persistent rockfall activity, the limestone cliffs of Lauterbrunnen appear rather compact and only marginally weathered. Thus, there is no reason to assume much lower values than those of 2000–3300 $ms^{-1}$ for S-waves in limestone from empiric tests (Bourbie et al., 1987). Accordingly, the first local maximum at lower velocities did not yield any consistent rockfall locations along the cliff, even when the other criteria clearly pointed at a rockfall. The average R$^2$ values for the higher velocity range from a broad plateau of equally likely velocities including 2700 $ms^{-1}$. Thus, based on information from both approaches, the average seismic wave

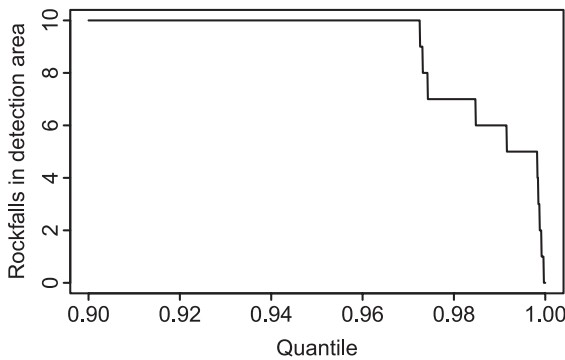

**Figure 8.** Number of rockfall trajectories inside location estimate polygons as function of minimum location estimate quantile.

velocity for running the location routine was set to 2700 $ms^{-1}$. Without the existence of independent locations of rockfall detachment zones, seismic velocity can only be constrained with low uncertainty by active seismics.

## 5.4 Location of rockfalls

The application of the location routine to the 37 potential rockfall events placed nine of them in the area of interest covered by our TLS surveys and the seismic network (Fig. 1). Eight further events were located along the west-facing valley side. Most of these had poor location constraints due to low SNR or inappropriate fits of the overall time delays of the signal envelopes. The other events could either only be located along the margins of the distance maps as the closest approximation for more distant sources, or were located west of the Lauterbrunnen Valley, higher in the catchment. One event, which showed all characteristics of a very proximal rockfall and subsequent rock avalanche but exhibited an extraordinarily wide frequency range (event 10 in Fig. 9) could successfully be located within the area of interest by manually setting the location frequency window to 5–35 Hz.

Thus, after extensive processing and manual verification, all ten TLS-detected rockfalls could be independently located by the seismic approach. SNRs of all ten events were above 5 and up to 59, depending on the magnitude of the event and the distance of the source to a seismic station. With the exception of the manually adjusted settings for event 10, the default settings resulted in an average difference between TLS (i.e., line of steepest descend from detachment zone) and seismic location of $81^{+59}_{-29}$ m. The maximum difference was 761 m (event 1, Table 1) because a significant part of the location estimate polygon for this event, including the location of $P_{max}$, was placed on the other valley side, separated from the cliff face by the entire valley floor. However, all TLS-based events were located within the default uncertainty areas defined by the 0.95 quantile, most of which were elongated by several 100 m in the north-south direction in plan view (Table 1). Some areas of uncertainty extend into the valley floor (events 6–8) but most were entirely within the cliff face. In five of the ten cases, $P_{max}$ is located higher on the cliff than the TLS-based detachment zones (i.e., events 1, 2, 3, 4, 9). We see the main causes for deviations in inhomogeneities of the solid media, resulting in spatially non-uniform seismic velocities. Specifically, there should be a velocity difference between the solid limestone that forms the cliff and the debris fabric that constitutes the talus slopes. Thus,

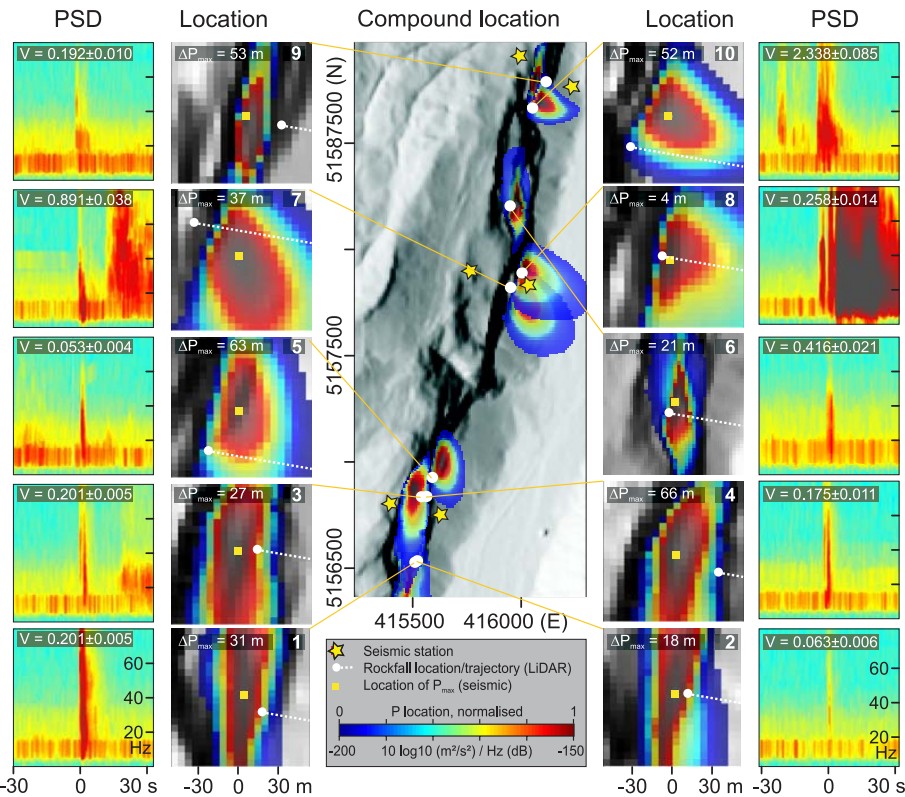

**Figure 9.** Seismic location of the 10 TLS-based rockfall events. Compound location map shows an overlay of all 10 detected events with coloured polygons corresponding to locations with cross-correlation values above the 0.973 quantile. Location close-up boxes are centred at $P_{max}$, i.e., the location with the highest cross-correlation value. PSD boxes show the spectral evolution of each event as recorded by station "Funny Rain". Event start is indicated by time zero. For event duration see Table 2. $\Delta P_{max}$ is the deviation of seismic location estimate from rockfall trajectory along steepest path. Locations of all rockfalls shown based on optimised location frequency windows for illustrative reasons (Table 1 for default deviations).

especially impact locations close to or at these talus slopes may be affected by larger deviations because the average seismic velocity successively fails to explain the arrival times of signals at the seismic stations.

Adjusting the frequency windows for the location routine to minimise the differences to the TLS data usually required shifts by less than 4 Hz. Events 7 and 8 required greater adjustments, as low-frequency windows yielded much better results (Table 2). Optimising the location settings resulted in average location differences of $33^{+20}_{-6}$ m with a maximum deviation of 66 m and a minimum deviation of 4 m.

Increasing the quantile thresholds to define the uncertainty polygons for each location estimate reduces their area, which eventually leads to a drop of the number of matches with TLS-based event location (Fig. 8). Up to a threshold value of 0.973, all ten rockfalls are included in the uncertainty areas.

## 6 Discussion

### 6.1 Rockfall detection from continuous seismic data

The challenge of detecting rockfalls with the seismic approach is to identify a few short target signals in month-long records of hundreds of samples per second. This is especially relevant for the small rockfall events of this study. Thus, the described routine for data processing is neither intended to be nor capable of coming close to automatic detection and location of rockfalls of this size. The workflow of signal processing and analysis significantly reduced the number of initially picked events by a factor of 4. This provided a reasonable base for the subsequent manual identification of likely rockfall events. The STA-LTA-ratio threshold values (i.e., 5 and 3) as well as the SNR threshold value (i.e., 6), determined from the two manually identified events in the control period, allowed detection of all ten rockfalls shown by the TLS data, even though all other events involved smaller volumes than the two manually identified ones. The initial filter frequency window for the STA-LTA-ratio approach of 10–30 Hz might have benefited from a lower cutoff frequency since some of the rockfalls showed optimal location frequencies well below this value (Table 1).

The monitored section of the Lauterbrunnen Valley is a comparably noisy environment. The example PSD (Fig. 5 a) shows ample signals from sources other than rockfall activity. A major source of falsely picked events was passing trains (87 %). For rockfalls as small as those detected in this study, raising the initial SNR threshold to exclude signals associated with train activity would result in rejecting most of the rockfall events. However, for rockfall volumes one or more orders of magnitude larger, this simple parameter adjustment should yield a significantly better detection result. The 19 detected earthquakes could have been removed based on differences in the relationships between magnitude, duration and frequency content (e.g., Manconi et al., 2016) or multivariate classification approaches (e.g., Provost et al., 2017). However, the duration distributions of rockfalls versus earthquakes already allowed a sufficient discrimination. Thus, although the data processing workflow is far from automatic and leaves one order of magnitude more events than the actual number determined from manual evaluation, it provides a systematic and reproducible way to detect rockfalls close to the lower limit of detection.

### 6.2 Rockfall location

All ten TLS-based rockfall events were confirmed with an average location error along the rockfall trajectory of 33 m when the frequency window of the location algorithm was adjusted manually. Without this optimisation, which is only possible when reference data are available, the location deviation was 81 m on average. This is comparable with errors of about 80 m from a rock avalanche study on Montserrat, Lesser Antilles, with a network of 11 stations (Levy et al., 2015). However, that study had a larger network aperture and focused on event volumes of $10^3$–$10^6$ m$^3$. Instead, rock mass volumes in the Lauterbrunnen Valley were generally well below 1 m$^3$ and our study had only four operating seismic stations, organised in a topology and station spacing that are comparable to those from other studies (Hibert et al., 2014; Burtin et al., 2016).

The TLS-based detachment locations and their rockfall trajectories are within the areas defined by the 0.95 quantile threshold (Fig. 9). Only when independent constraints on the location of the seismically recorded events are available, is it possible to investigate the validity and effectiveness of this arbitrary threshold. In this study area, the threshold can be increased up to

0.973 to still provide a valid uncertainty estimate for possible rockfall locations/trajectories. Effectively, this means that the area of each uncertainty polygon can be decreased by 45 %.

An important issue is that for some rockfalls the best location estimate ($P_{max}$) is above the actual rockfall detachment zone. This may be related to the extreme topography of the Lauterbrunnen Valley. The studied rockwall is up to 800 m high, yet it is represented by as little as four plan view pixels in the 10 m DEM and distance maps (cf. ranges of $z_{seis}$ in Table 1). Arguably, the lateral offset of rockfall location $P_{max}$ from the line of steepest descend is more important from a hazards point of view.

Assigning the locations of the ten seismically detected rockfalls to those detected by TLS is unambiguous in most cases. However, rockfalls with comparable volumes from similar detachment heights can be hard to distinguish. For example, events 3 and 4 are located 44 m apart, at 1108 and 1064 m asl., and released 0.201 and 0.175 m$^3$ of rock, respectively. Accordingly, their seismic waveforms and PSDs (Fig. 9) look very similar and there remains ambiguity about the seismic identification as stated in Table 1. This has consequences for the temporal information associated with the seismic data. But in this case, both events occurred on 20 October, one at 3 pm, the other at 7 pm. Ambiguity also arises for events 1 and 2. However, there the rockfall volumes allow for a better matching with the seismic results. Event 1 entrained 0.201 m$^3$ whereas event 2 mobilised only 0.063 m$^3$ from a near identical position and fall height. Accordingly, the emitted seismic energy of event 1 should be significantly higher than event 2, which is reflected in the corresponding PSD, where event 1 shows a much longer and more powerful signal. Hence, if the geometric properties of the released rock masses are sufficiently distinct, it is possible to disentangle nearby events from the detailed seismic information.

For large ($> 10^4$ m$^3$) gravitational mass wasting processes there appear to be robust relationships between released volumes and a series of seismic attributes (Dammeier et al., 2011; Ekström and Stark, 2013). However, such large events affect significant areas, even entire slopes. In contrast, the small volumes mobilised in the Lauterbrunnen Valley do not show such clear volume-based relationships (apart from the one example described above). The largest event (2.338 m$^3$) did not yield the highest signal intensities or longest duration, and vice versa for the smaller events. The combination of released volume, detachment height above cliff base, the number, distance and strength of intermediate impacts, the degree of fragmentation during the fall phase and the fate to the rock mass on the talus slope (direct deposition, subsequent downhill translocation, entrainment of impacted talus) resulted in a polymorphic seismic signal, which complicates direct links of seismic parameters with geometric or kinetic properties of the detected rockfalls at this spatial scale. To explore such questions about relations among volume, detachment height, fragmentation and debris entrainment upon impact – all obviously more useful for larger rock volumes than found in this study – the combination of TLS and seismic monitoring provides all necessary sources of information. The high temporal resolution and ability to detect small volumes makes especially the seismic technique perticularly interesting for studies of relations between rockfalls and environmental conditions that are suspected to cause them (Dietze et al., 2017).

The apparent seismic detection limit for rockfall volumes in the Lauterbrunnen Valley is well below 1 m$^3$. This is remarkable given that the stations are mostly more than one km apart and that most of the rockfalls used for validation originated at the lower cliff parts, resulting in limited kinetic energy upon impact. Location feasibility is however not only determined by the rockfall volume and drop height. The distance between impact location and location of the seismic stations, the inelastic attenuation properties of the rock and the energy dissipation due to rock fragmentation (e.g., Hibert et al., 2011) also determine

the potential to successfully locate the rockfall. The possibility to analyse rockfalls as small as 0.053 m³, impacting at distances of 170–1950 m from the seismic stations, makes seismic monitoring a method that is able to reveal events well below the resolution of most other post-event survey techniques, such as aerial and satellite imagery analysis.

Unlike other rockfall survey techniques, seismic methods allow for monitoring of rockfalls with high temporal resolution, down to fractions of a second. During the first half of the monitored month only two rock masses were released, while the other half of the month saw the majority of events. Beyond this, the high temporal resolution allows connecting the events to ambient conditions and trigger mechanisms, and to study process interactions (e.g., Helmstetter and Garambois, 2010; Burtin et al., 2014; Zimmer and Sitar, 2015).

### 6.3 Rockfall anatomies

Seismic monitoring allows detailed insight into the dynamics of rockfalls. The exemplary event (Fig. 2) consisted of three distinct phases and lasts in total for almost a minute. Phase 1 (less than one s duration) is the first notable seismic activity after minutes of calm at all stations. It reflects the seismic signal associated with initiation of the rockfall event. The high frequency content may either correspond to the rebound of the cliff after detachment of the mass (e.g., Hibert et al., 2011) or the opening and propagation of fractures rather than impacts of a moving rock mass. The latter interpretation is supported by seismic records from the Illgraben, Rhone Valley, Switzerland, that show an exponentially increasing density of signals, which indicate cracking or fracture propagation (Zeckra et al., 2015) starting days before a $10^4$ m³ large rock avalanche took place (Burtin et al., 2016). The spectral properties of these signals (short, less than 1 s pulses at 20–50 Hz), recorded by a seismic station about 150 m away from the initiation zone of the rockslide are very similar to the first phase of the rockfall from the Lauterbrunnen Valley (Fig. 2).

Phase 2 (one s duration) begins 1.7 s after this fracture propagation phase and may reflect the impact of the released rock mass on the cliff face. The predominantly low frequency content implies that the mass is still intact upon the first collision. Low frequencies can only be generated by large rock masses that convey a high momentum rather than a series of smaller particles hitting a surface simultaneously (Burtin et al., 2016). The strong impact likely caused fragmentation of the rock, because there is no low frequency content in any of the later signals from this event. The rock fragments experienced a free fall phase (calm period in all signal waveforms) of approximately 7.5 s, corresponding to a drop height of 271 m. With a detachment elevation between 1048–1123 m asl. this places the impact somewhere in the central part of the talus slope that reaches from 910 m asl. at the cliff base to 820 m asl. on the valley floor.

Phase 3 (about 40 s duration) represents the continuous impact of the fragmented rock mass on the talus slope for tens of seconds. This activity very likely graded into a phase of downslope translocation of debris and entrainment of further talus. The PSD of phase 3 shows the typical properties characteristic of rock avalanches (e.g., Suriñach et al., 2005).

Similar insights to the anatomy of events are possible for the remaining nine rockfalls, though often with less rich detail or variability. Readers are invited to explore the data contained in the supplementary materials. This one anatomy of an example event highlights the universality of seismic sensors to investigate the dynamics of a rapid mass wasting process at a level of detail that would otherwise require an expensive and time-consuming multi-sensor approach, consisting of, for example, video

imagery, prior and a posteriori TLS scans, perhaps further acoustic sensors, and post-event field mapping. Furthermore, the area of interest can only be small to be covered by these alternative techniques. Thus, the installation must be placed at "the right spot", instead of relying on the flexibility to monitor a wider area with a seismic network.

## 7 Conclusions

The detachment locations of ten rockfall events, as small as $0.053 \pm 0.004\,\mathrm{m}^3$ and totaling a volume of $4.789 \pm 0.100\,\mathrm{m}^3$, were detected by TLS over 37 days. Using broadband seismometers, these events were independently detected and located with an average deviation of $81^{+59}_{-29}$ m. Further seismic rockfall signals were detected and located outside this instrumented cliff area. The seismic signatures allow i) insight into the dynamics of single events, ii) quantification of the exact event onset time and duration, and iii) calculating minimum fall heights. It is thus possible to monitor rockfalls sensu stricto with a significant free

fall phase and a pronounced short impact phase. This extends the previous field of applications of environmental seismology to more extreme settings. Our results suggest that seismic monitoring with a network geometry comparable to other natural scale experiments (e.g., Lacroix and Helmstetter, 2011; Burtin et al., 2014) is a valid approach to catchment-wide detection, location and characterisation of Earth surface activity in an exceptionally steep terrain. Our data complements work that has focused on the coupling of rockfall to other processes in the sediment cascade of mountainous landscapes (e.g., Krautblatter et al., 2012).

Further advantages of seismic rockfall monitoring are i) its ability to detect events independent of visibility conditions, a major limitation of remote sensing techniques, including time lapse camera surveys, and ii) the large size of the area that can be monitored with a limited number of stations.

The main limitations of this approach include that estimates of rock volume based on the emitted seismic energy or peak ground acceleration were not possible for the small rockfall events identified in this study. This was mainly due to the influence

of intrinsic factors, such as the proportion of energy consumed for fragmentation during the event or contribution of mobilised debris to the seismic signals upon impact on the talus slope. A further challenge is the high effort due to manual removal of false events under such conditions. This drawback represents a serious issue when attempting fully automated approaches of rockfall detection.

While the combined description of event location and precise timing information of rockfall activity would provide access

to trigger mechanism analysis in principle (Dietze et al., 2017), the number of detected events from this study is too small for this goal. At larger scales (regarding released volumes and monitored area) there are first-order effects that allow relating seismic metrics to process parameters (e.g., Hibert et al., 2011; Dammeier et al., 2011; Ekström and Stark, 2013). Thus, when increasing the monitored area and focusing on larger released volumes ($> 10^4\,\mathrm{m}^3$), environmental seismology could become a real alternative to classic rockfall observatory instruments, with the capability to go beyond these by simultaneously recording

proxies of environmental triggers and resolving process coupling and interaction.

There is a potential to optimise the parameters for event location but there is no straightforward way to do this without independent auxiliary information. Hence, a realistic location error range along the trajectory of released rocks is 52–140 m (interquartile range). The height and location of the detachment zone can only be provided by seismic methods if the detach-

ment process can be recorded and the subsequent impacts of the released rock mass can be located with sufficient confidence to allow back-calculation of the falling time. Rockfall release zones that are separated below the level of seismic location confidence can be deciphered from each other if the released volumes are different from each other and generate sufficiently distinct seismic characteristics. Hence, combining seismic and TLS methods can provide a very detailed complementary picture of rockfall activity.

## 8  Data and code availability

The seismic data used in this study is available in the supplementary materials, along with a detailed documentation about how to use it to reproduce the results of this study. The digital elevation model data set cannot be made freely available, but may be replaced by equivalent data to reproduce the results. TLS point cloud data are available upon request.

*Author contributions.* Michael Dietze contributed to seismic fieldwork and data analysis. Solmaz Mohadjer contributed to TLS fieldwork and data processing. Jens M. Turowski, Todd A. Ehlers and Niels Hovius contributed to equipment provision, project planning and data analysis. All authors contributed to manuscript preparation.

*Acknowledgements.* The field work campaigns generously benefited from the support of Maggi Fuchs, Michael Krautblatter, Torsten Queißer and Fritz Haubold. The authors are thankful for these creative involvements. We further thank Josy Burke and Lorenz Michel for TLS field assistance. We are also thankful to the GIPP seismic device pool for providing six TC120s sensors and Cube[3] data loggers. We also thank Valerie Zimmer, Clément Hibert, Agnès Helmstetter, an anonymous referee and the Associate Editor for their input to improve previous versions of the article.

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
