# Peer review of "Seismic monitoring of small alpine rockfalls – validity, precision and limitations"

_Earth Surface Dynamics, 2017_

## Referee Comment (RC1) · V.L. Zimmer (Referee) · 11 Apr 2017

Main Text:

The detection of extremely small events is impressive and exciting. Good work on the locations - this is admitted difficult with a fractured surface and a moving source.

P. 3 Line 16: Describe the characteristic frequency content in more detail (there is some discussion later, this may suffice, but I was wondering about it early on)

P. 6 Line 29: Zimmer and Sitar 2015 may be a better reference for this close-range work (<1km) than Zimmer 2012 (at 6+ km) - V.L. Zimmer, N. Sitar / Engineering Geology 193 (2015) 49–60

P. 7 Line 16-17: "signal, i.e., several seconds rise time of the signal from background,

followed by a long decay into background noise after reaching a maximum amplitude" The 2012 paper dealt with a very large event at distance: smaller events did not always have the same signal (see Zimmer and Sitar 2015)

P. 7 Lines 30-35: It's a little unclear how this method works, especially with a moving source - as you note below, the rocks are moving roughly vertically down the slope. It might be interesting in a future work to use this method for segments of the single and create a set of temporal correlations that together show the path? I'm not sure how this would work, but it would be interesting?

Table 1 - delta Pmax pre and post optimization: I'm not sure I understand what happens during optimization. Figure 7 looks like you have great location results, but this table seems to show that there was an iterative processing that eliminated some seismic locations and chose others based on additional evidence (e.g. TLS). Can you elaborate or make this a bit clearer? (this is explained later in the text)

P. 11 Lines 9: Is the 5-15 Hz (progressively decreasing signal) correlated with river baseline flow patterns (e.g. fluctuations in river flow due to increased evapotranspiration during the daytime?) Correlations of seismic/acoustic noise with river flow is hard to do, so it would be an interesting result if your instruments were sensitive to this.

P. 11 Line 14: I love this "Swiss trains always run on time". However, always is misspelled "alyways".

P. 11-13 and Figure 4 (environmental noise): I think this is one of the more interesting challenges to discriminating rockfalls from other sources (working toward automatic event detection). Good job identifying other sources and applying rejection criteria (including multi-station detection) to get down to 500 events, and then further confirming the identity of many of those spurious events.

Section 5.3 and 5.4: It took a bit of reading to understand the velocity estimate and the location methodology. The assumption is that the waves are arriving on a relatively

direct path, but if they are not (if there is weathering, sheeting joints, and variable velocities), it may explain some of the location deviation from seismic-only methods. (Also the valley floor probably has a substantially lower velocity - accounting for that may significantly shrink your location polygons in the cases when Pmax extends to the valley floor- e.g. event 1 and events 6-8). Nevertheless, the fact that you were able to localize such small and mobile seismic sources to the degree that you did is impressive.

P. 19 Line 18: could some of the shift to lower frequencies be attributed to overall lower energy release and higher proportional attenuation of higher frequency signals? E.g. with less energy, the high frequency portion of the signal is too low to be detected above ambient noise?

P. 20 Line 17: sensu stricto, not sensu strictu?

Supplemental material: I tested the supplemental material, and ultimately got it working on my Windows machine (but not my old Mac). Except for the DEM portion (I did not download a DEM), it all worked as promised. Some minor challenges that I noted:

Eseis needs: Version 3.3 of R or later (I have an old computer which can't use Version 3.3, are there older versions?)

Eseis on Windows needs to be installed from your webpage - some of the compiled files aren't readable by Windows, but I did manage to make it all work!

Eseis on Windows also needs Rtools

On Mac, (Error: Don't know how to decompress files with extension 17.1), might be related to Version 3.2 of R (unresolved)

What data format and resolution does the DEM need to be in? USGS DEM raster?

---

## Author Comment (AC1) · 13 Apr 2017

article pdfpages hyperref listings

We would like to thank the referee for the encouraging and helpful comments, all of them obviously devoted to improve the quality and impact of the manuscript.
* * *
*Reviewer 1.1*: *The detection of extremely small events is impressive and exciting. Good work on the locations - this is admitted difficult with a fractured surface and a moving source.*

**Reply**: We are glad to see the positive impression our initial version of the manuscript made.
* * *
*Reviewer 1.2: P. 3 Line 16: Describe the characteristic frequency content in more detail (there is some discussion later, this may suffice, but I was wondering about it early on)*

**Reply**: Details given in future revised manuscript version (P.3, Line 16–17), with link to our example in figure 2 b and a reference from another study.
* * *
*Reviewer 1.3: P. 6 Line 29: Zimmer and Sitar 2015 may be a better reference for this close-range work (<1km) than Zimmer 2012 (at 6+ km) - V.L. Zimmer, N. Sitar / Engineering Geology 193 (2015) 49–60*

**Reply**: Indeed, the suggested reference is more appropriate and has been used in the revised version.
* * *
*Reviewer 1.4: P. 7 Line 16-17: "signal, i.e., several seconds rise time of the signal from background followed by a long decay into background noise after reaching a maximum amplitude" The 2012 paper dealt with a very large event at distance: smaller events did not always have the same signal (see Zimmer and Sitar 2015)*

**Reply**: The sentence has been corrected, the suggested reference (Zimmer and Sitar, 2015) is placed after mentioning the other mode of rockfall signals: "more erratic peaks in the seismogram as the result of impulsive impacts".

**ESurfD**
[Figure]

*Reviewer 1.5: P. 7 Lines 30-35: It's a little unclear how this method works, especially with a moving source - as you note below, the rocks are moving roughly vertically down the slope. It might be interesting in a future work to use this method for segments of the single and create a set of temporal correlations that together show the path? I'm not sure how this would work, but it would be interesting?*

**Reply**: Indeed, the location must be estimated for each impact individually to minimise smearing effects of a moving source. This is now mentioned in the text. Actually, the first author has submitted a further manuscript that explicitly shows how the method allows locating successive impacts of a complex rockfall sequence: http://www.earth-surf-dynam-discuss.net/esurf-2017-20/ (see figure 3).

*Reviewer 1.6: Table 1 - delta Pmax pre and post optimization: I'm not sure I understand what happens during optimization. Figure 7 looks like you have great location results, but this table seems to show that there was an iterative processing that eliminated some seismic locations and chose others based on additional evidence (e.g. TLS). Can you elaborate or make this a bit clearer? (this is explained later in the text)*

**Reply**: The description of the table is now improved: deviations with the default location frequency windows are now shown first, the ones with optimised frequency windows in brackets. Also, it is now noted that optimisation is only possible when independent information on rockfall locations is available, as was mentioned in the text further down, already. Additionally, figure 7 was updated. We changed two outdated location deviation values due to older calculations (31 instead of 37 m for event 1, 21 instead of 26 m for event 6). The figure caption does now explicitly mention that the location estimates are shown for optimised location frequency windows for illustrative reasons. In fact the

difference would have been obviously visible only for events 1 and 5.
* * *
*Reviewer 1.7: P. 11 Lines 9: Is the 5-15 Hz (progressively decreasing signal) corre-lated with river baseline flow patterns (e.g. fluctuations in river flow due to increased evapotranspiration during the daytime?) Correlations of seismic/acoustic noise with river flow is hard to do, so it would be an interesting result if your instruments were sensitive to this.*

**Reply**: We have added a reference to the data source (runoff of the main river draining the valley) and an article providing background and agreement of the frequency pattern with our data (Gimbert et al., 2014). Indeed, the seismic signature of this frequency band shows a clear link to the runoff patters of the river, but this topic is a bit out of scope for the manuscript, though still valuable.
* * *
*Reviewer 1.8: P. 11 Line 14: I love this "Swiss trains always run on time". However, always is mis-spelled "alyways".*

**Reply**: We also love the term, and have corrected the typo.
* * *
*Reviewer 1.9: P. 11-13 and Figure 4 (environmental noise): I think this is one of the more interesting challenges to discriminating rockfalls from other sources (working to-ward automatic event detection). Good job identifying other sources and applying re-jection criteria (including multi-station detection) to get down to 500 events, and then further confirming the identity of many of those spurious events.*

**Reply**: Indeed, the period of manual event filtering was an interesting exercise to learn patience. We have added the referees comment (identifying such small events is a challenge on the way to fully automatic detection systems) to the conclusions.
* * *
*Reviewer 1.10: Section 5.3 and 5.4: It took a bit of reading to understand the velocity estimate and the location methodology. The assumption is that the waves are arriving on a relatively direct path, but if they are not (if there is weathering, sheeting joints, and variable velocities), it may explain some of the location deviation from seismic-only methods. (Also the valley floor probably has a substantially lower velocity - accounting for that may significantly shrink your location polygons in the cases when Pmax extends to the valley floor- e.g. event 1 and events 6-8). Nevertheless, the fact that you were able to localize such small and mobile seismic sources to the degree that you did is impressive.*

**Reply**: The referee is correct, there are a series of modifications of the ideal medium under natural conditions. We discuss these potential causes for decreased P$_{max}$ values now chapter 6.2, first paragraph.
* * *
*Reviewer 1.11: P. 19 Line 18: could some of the shift to lower frequencies be attributed to overall lower energy release and higher proportional attenuation of higher frequency signals? E.g. with less energy, the high frequency portion of the signal is too low to be detected above ambient noise?*

**Reply**: Indeed, we have added this alternative explanation to the discussion of this rockfall phase.
* * *
*Reviewer 1.12*: *P. 20 Line 17: sensu stricto, not sensu strictu?*

**Reply**: We changed the typo.
* * *
*Reviewer 1.13*: *Supplemental material: I tested the supplemental material, and ulti-mately got it working on my Windows machine (but not my old Mac). Except for the DEM portion (I did not download a DEM), it all worked as promised. Some minor chal-lenges that I noted: Eseis needs: Version 3.3 of R or later (I have an old computer which can't use Version*

**Reply**: We thank the referee for also checking – and independently confirming the analysis by working with – the supplementary material. Indeed, the R package 'eseis' requires at least R version 3.3. This is due to the dependencies of the package, i.e., other packages that 'eseis' uses, and thus not in our hands. The first author of this manuscript just recently submitted another manuscript to the journal Ancient TL in which exactly this drawback of quickly evolving software is stressed. There are further shortcomings related to working with the package under other than Unix systems, as mentioned below. All these shall however be obsolete once the package is built for CRAN, the comprehensive R Archive Network, a step that is envisioned for the near future, once the pace of added functionality has decreased to a manageable level (cf. http://www.micha-dietze.de/videos/eseis_history.avi).

In summary, the first author is very delighted to see that curious and eager researchers indeed attempt and succeed to reproduce the results of the manuscript with the raw data published along with the manuscript, using free and open software. The first author sees this as a perfect example of open and transparent science, despite the efforts and pitfalls associated with going this way.

*Reviewer 1.14: 3.3, are there older versions?) Eseis on Windows needs to be installed from your webpage - some of the compiled files aren't readable by Windows, but I did manage to make it all work!*

**Reply**: Correct, the Github repository of the package and the convenient functionality of the 'devtools' package to install 'eseis' from Github is not available for Windows platforms by default. On my website there is a note and link that help one to install the package 'devtools' on any of the common computer platforms.

*Reviewer 1.15: Eseis on Windows also needs Rtools On Mac, (Error: Don't know how to decompress files with extension 17.1), might be related to Version 3.2 of R (unresolved)*

**Reply**: Indeed, running R version 3.3 is essential for installing and using 'eseis', which is in turn required to locate the events depicted by the raw seismic data. As noted above, there are not many alternatives to this (such as sourcing all relevant functions of the package source code by hand). The easiest solution would probably be to install a proper Linux system in a virtual box and therein an up to date version of R.

*Reviewer 1.16: What data format and resolution does the DEM need to be in? USGS DEM raster*

**Reply**: The package in its latest version (already 0.3.2 now) has been revised and additional comments about supported file formates as well as projection constraints are added. The supplementary material has also been updated concerning the DEM file

formats and metric units constraint. In general, any meaningful resolution of the DEM is suitable, depending on the relative size of the area of interest and target resolution.

---

## Referee Comment (RC2) · C. Hibert (Referee) · 27 Apr 2017

Dear Authors, dear Editor,

This study focuses on the seismic detection and location of small rockfalls that originated from an unstable cliff of the Lauterbrunnen Valley in the Swiss Alps. The Authors show that they are able to accurately locate some of the rockfalls they detected. They also show that some qualitative information on the dynamics of the events can be inferred from the recorded seismic signals.

Using seismology to detect and study rockfalls, and more generally gravitational instabilities, is challenging but constitutes a high research priority to improve the completeness of catalogs and the assessment of hazards associated with slope failures. In this context, I think this study can become an excellent contribution to promote the use

of seismology for rockfalls studies. I found the manuscript clear and well-written but I identified several minor issues that have to be addressed. I also have few suggestions that might improve the overall impact of this study. Below are my comments and those suggestions.

Best regards,

Clément Hibert

Comments and suggestions:

Title: When first reading the title of the paper I was expecting an analysis of the feasibility to study rockfalls with seismic methods in different contexts, or an enriched review of past studies on this subject. 'Validity' and 'limitations' of seismic rockfalls monitoring in general are not discussed in this paper and I find that what the authors propose is essentially an interesting case study. This should be explicit and clarified in the title. I would suggest for example: "Validity, precision and limitations of the seismic detection and location of small rockfalls in the Swiss Alps".

P2 L17: Please order the references chronologically. There is a wealth of studies on landslides seismic signals. If you decide to select some of them as examples, use "e.g." before citing them.

P2 L18: While it would be an honor to share the name of David Hilbert, I am not. Here and everywhere else please correct the references to "Hibert et al." (no "L").

P2 L21: "cf" not necessary here or elsewhere.

P3 L14: Is this different from spectrograms?

P3 L19: Burtin et al. [2016] were not the first to show that seismic signals generated by rockfalls are dominated by surface waves. In the references here you can add Deparis et al., [2008], Dammeier et al., [2011], and Levy et al., [2015].

P3 L22: "Vital" seems a bit strong. Interesting? Significant? Crucial?

Section 3: Is this section necessary? Could you move this into the introduction?

P3 L28: What is the "limit of detection"? Is it the targeted (or the possible?) resolution of the point clouds?

P4 L3: Are the seismometers 3 components? Please add this information here.

Figure 2 – caption: Do you know the volume of this particular event? If so this information could be added in the caption. Are the signals filtered?

P6 L17: The STA-LTA ratio picker was first proposed by Allen [1982].

P6 L18: Envelopes of seismic signals are commonly computed from the Hilbert transform of the signal. I think using the absolute amplitude is not a problem for detection, but for a localisation method based on the cross-correlation of envelopes, the Hilbert transform might yield better results. While I certainly do not think it is necessary to redo the current analysis with Hilbert envelopes, I would suggest testing this in future studies.

P6 L24: Can you indicate here what are the threshold values chosen?

P7 L1 and 11: You choose here a velocity for S-waves but as stated before rockfalls seismic signals are dominated by surfaces waves (which are slower than body waves). How many events have you excluded based on this criterion?

P7 L10-11: This is a bit confusing. You first had an automated exclusion criterion based on the time delay between the onsets of the waves recorded at each station of the network and then you still check manually if this criterion is verified? What is the point of the first automated exclusion then? Maybe reorganize this paragraph and the one just before to improve clarity.

P7 L13-14: Criterion iv) : Does this imply that you know the location of the events before manually selecting the signals?

P7 L17: References : "e.g." or add at least Surinach et al. [2005].

P7 L20: "Multiptaper" Typo?

P7 L21: References : "e.g.". Hibert et al. [2011] and Dammeier et al. [2011] seem more appropriate references here.

P7 L29-34: The approach proposed by Hibert et al. [2014 and not 2011] is designed to overcome all the issues regarding the specificity of rockfall seismic signals you enumerate before this sentence (emergent onset, waveform discrepancies, absence of seismic phases, high-frequency). Moreover if the dominant issue is the "differences between waveform properties at different stations" the cross-correlation approach would not work. In your case you can use a method based on the cross-correlation of the signal waveforms because the signals recorded at different stations are not too dissimilar. I suspect this is the case because the aperture of your network is not large (inter-station distances of ~1km). This is not the case at the Piton de la Fournaise volcano and is one of the difficulties that forced us to develop a new kurtosis-based first-arrival picker that is accurate enough to pick emergent signals. Please rewrite this paragraph by taking this remark into account.

P8 L19-21: Topography correction is necessary because rockfalls generate surface waves that propagate following the topography. Also the correct reference is Hibert et al. [2014] and not [2011] here.

P8 L8-18: If you change the frequency range used to find the time lag that yields the best cross-correlations this should have an impact on the optimal velocity, and vice-versa. Can you elaborate on the interdependence of the optimal frequency bands and the optimal velocities found? It can be interesting to add in Table 2 the velocity that gives the best location for these 10 rockfalls.

P8 L21-22: Not all Earth surface processes generate seismic signal dominated by surface waves. I suggest to change "other Earth surface processes" into "other mass-movement processes" or "gravitational processes". Also see comment on P3 L19 regarding the reference to Burtin et al., [2016].

P8 L22-25: It is not clear how you performed this correction. What is: "that part where direct distance is above the actual surface elevation"? Does this mean that you corrected the direct straight line distance from pixel to pixel by the slope angle? Did you compute profiles for each pixel-station pair from the intersection of the straight line between those two points with the grid points of your DEM? Integrating the topography in propagation maps is not a trivial task but as you mentioned is critical to have accurate locations of rockfalls. This should be a bit more detailed, especially if the main focus of this paper is the capability to locate rockfalls from the seismic signal they generate.

P8 L30: Please define what is the "likelihood quantile" before.

P9 L26: What caused this tilting? Do you know when it started? What is the influence of this tilting on the seismic signals recorded before dismantling those stations?

P11 L16-18: I am a bit sceptical regarding the "rain drop sources" because you have buried your stations at 30-40 cm depth. This should prevent any direct contact between the seismometers and rain drops and I think rain drops are too weak seismic source to generate signals that would not be attenuated in the first few centimeters of propagation. Other common sources that can generate impulsive signals with energy in high-frequency bands are thunder, numerical glitches or close footsteps (animal or human). You based your attribution of those signal to rain drops from the observation that "it only occurs in the records when it was raining in the Lauterbrunnen Valley during deployment and maintenance of stations". So you observe these noise signals on the days you were on the sites. There is a possibility that these signals are your footsteps, but without clear evidences we do not know. So did you observed those signals on days where you were not on site? Can you provide other arguments to attribute those signals to rain drops? For example, did you observe that those signals appeared and disappeared gradually over a period of times of several tens of minutes (or few hours), mimicking the passing of scattered showers? If so could you show this on a figure to definitely convince your readers that those signals are indeed generated by rain drops? If not I would suggest to rename this class of source to "impulsive noise".

P14 L3-4: What is "n"? What is "r"?

P14 L4: If you want to provide to the readers an analysis based on the SNR you need to indicate how you have computed this quantity before.

P17 L13-14: What are those relationships? Do you refer to the studies of Hibert et al. [2014], Manconi et al. [2016]? Dammeier et al. [2016]?

P17 L20-21: Levy et al. [2015] used a different approach (first-arrival picking and not cross-correlation back propagation as in the present study) and had a network with a much larger aperture (inter-stations distances of few kilometers).

P18 L7-9: I agree with the assumption that a rockfall with a higher volume should generate a higher-amplitude seismic signal if the travelled path and the fall height are the same. However you say latter that in your case there are no correlation between seismic energy/amplitude and the volumes of the events.

P18 L12-13: The volume of the rockfalls in the study by Hibert et al. [2011] had volumes as low as few cubic meters.

P18 L18-19: Indeed, you are working with complicated events and I acknowledge that extracting quantitative laws might be difficult in this case. However, as shown by the example discussed in section 6.3, you are able to identify the different stages of the rockfall propagation. Is this true for the 10 rockfalls in your database? If so, you have every information you need (location of the events, volumes/masses, average velocity of the medium) to go further in your analysis. For example, what are the relationships between the first impulsive arrival amplitude (corrected from propagation effect) (phase 1) and the volume? The relationships between the seismic energy and the potential energy lost at the first impact with the topography (phase 2) ? The same relationships during the propagation phase on the talus (phase 3)?

Those are fundamental issues that you might be able to contribute to answer with your dataset. Even if no relationships are found, this would still be very interesting as it will

nourish discussion on the validity for small rockfalls of the relationships found by others [e.g. Deparis et al., 2008; Vilajosana et al., 2008; Hibert et al., 2011; Dammeier et al., 2011; Yamada et al., 2012; Ekström and Stark, 2013; Farin et al., 2015; Levy et al., 2015; Hibert et al., E-Surf- in press]. I understand that this might be out of the scope of this study, but I think that adding this deeper analysis will significantly improve the impact and the reach of your paper.

P18 L31: references : add "e.g." and/or other references, for example : Helmstetter and Garambois [2010], Yamada et al., [2012], Zimmer & Sitar., [2015], Hibert et al. [2017].

P19 L3-5: While it seems reasonable to think that large mass detachments are preceded by cracking and fracture opening that generates an increasing rate of microearthquakes, this is more debatable for very small rockfalls such as the ones in this study. Another assumption to explain this first impulsive signal is that it is generated by the rebound of the Earth in the departure zone due to the detachment of the mass. This was observed at Piton de la Fournaise volcano [Hibert et al., 2011]. I think both assumptions should be mentioned here.

P19 16-17: If larger particles have higher momentum they will reach the bottom of the slope more rapidly than small particles. In fact this is what is observed in many cases on video recordings of events: large blocks preceding the flow of small granular materials. The loss of high-frequency at the end of seismic signals generated by gravitational instabilities is complex and still not yet fully understood. Analytical models [e.g. Okal, 1990; Farin et al., 2015] suggest that events with larger volume will indeed generate signal with a lower corner frequency, but the overall amplitude of the signal across the whole frequency range will be higher. To this adds the fact that high-frequencies generated by small particles are more attenuated. The combination of those two processes suggests that the loss of high-frequency at the end of those seismic signals is due to the early immobilization of the largest particles, not the smallest. This is highly speculative, and any interpretation of this frequency shift has to be done carefully and

**ESurfD**
supported by data. If you want to comment on this, please add references.

References cited in this review:

Allen, R., (1982), Automatic phase pickers: Their present use and future prospects. Bulletin of the Seismological Society of America, 72(6B), S225-S242.

Dammeier, F., Moore, J. R., Haslinger, F., & Loew, S. (2011). Characterization of alpine rockslides using statistical analysis of seismic signals. Journal of Geophysical Research: Earth Surface, 116(F4).

Dammeier, F., Moore, J. R., Hammer, C., Haslinger, F., & Loew, S. (2016). Automatic detection of alpine rockslides in continuous seismic data using hidden Markov models. Journal of Geophysical Research: Earth Surface, 121(2), 351-371.

Deparis, J., D. Jongmans, F. Cotton, L. Baillet, F. Thouvenot, and D. Hantz (2008), Analysis of rock‐fall and rock‐fall avalanche seismograms in the French Alps, Bull. Seismol. Soc. Am., 98(4), 1781–1796,

Ekström, G., & Stark, C. P. (2013). Simple scaling of catastrophic landslide dynamics. Science, 339(6126), 1416-1419.

Farin, M., Mangeney, A., Toussaint, R., Rosny, J. D., Shapiro, N., Dewez, T., ... & Berger, F. (2015). Characterization of rockfalls from seismic signal: insights from laboratory experiments. Journal of Geophysical Research: Solid Earth, 120(10), 7102-7137.

Helmstetter, A., & Garambois, S. (2010). Seismic monitoring of Séchilienne rockslide (French Alps): Analysis of seismic signals and their correlation with rainfalls. Journal of Geophysical Research: Earth Surface, 115(F3).

Hibert, C., A. Mangeney , G. Grandjean and N.M. Shapiro (2011). Slopes instabilities in the Dolomieu crater, la Réunion island : from the seismic signal to the rockfalls characteristics. J. of Geoph. Res., 116.
Hibert, C., A. Mangeney, G. Grandjean, C. Baillard, D. Rivet, W. Crawford, N.M. Shapiro, C. Satriano, A. Maggi, P. Boissier and V. Ferrazzini (2014). Automatic identiïfication, location and volume estimation of rockfall at Piton de la Fournaise volcano. JGR – Earth Surface . 119, 1082–1105.

Hibert, C., Mangeney, A., Grandjean, G., Peltier, A., DiMuro, A., Shapiro, N. M., . . . & Kowalski, P. (2017). Spatio-temporal evolution of rockfall activity from 2007 to 2011 at the Piton de la Fournaise volcano inferred from seismic data. Journal of Volcanology and Geothermal Research.

Hibert, C., J.-P. Malet, F. Bourrier, F. Provost, F. Berger, P. Bornemann, P.Tardif, and E. Mermin, Single-block rockfall dynamics inferred from seismic signal analysis, E-Surf, in press.

Levy, C., Mangeney, A., Bonilla, F., Hibert, C., Calder, E. S., & Smith, P. J. (2015). Friction weakening in granular flows deduced from seismic records at the Soufrière Hills Volcano, Montserrat. Journal of Geophysical Research: Solid Earth, 120(11), 7536-7557.

Manconi, A., Picozzi, M., Coviello, V., De Santis, F., & Elia, L. (2016). Real‐time detection, location, and characterization of rockslides using broadband regional seismic networks. Geophysical Research Letters, 43(13), 6960-6967.

Okal, E. A. (1990). Single forces and double-couples: a theoretical review of their relative efficiency for the excitation of seismic and tsunami waves. Journal of Physics of the Earth, 38(6), 445-474.

Suriñach, E., I. Vilajosana, G. Khazaradze, B. Biescas, G. Furdada, and J. M. Vilaplana (2005), Seismic detection and characterization of landslides and other mass movements. Nat. Hazards Earth Syst. Sci., 5(6), 791–798, doi:10.5194/nhess-5-791-2005.

Vilajosana, I., Suriñach, E., Abellán, A., Khazaradze, G., Garcia, D., & Llosa, J. (2008).

Rockfall induced seismic signals: case study in Montserrat, Catalonia. Natural Hazards and Earth System Science, 8(4), 805-812.

Yamada, M., Y. Matsushi, M. Chigira, and J. Mori (2012), Seismic recordings of landslides caused by Typhoon Talas (2011), Japan, Geophys. Res. Lett., 39, L13,301, doi:10.1029/2012GL052174.

Zimmer, V. L., & Sitar, N. (2015). Detection and location of rock falls using seismic and infrasound sensors. Engineering Geology, 193, 49-60.

Please also note the supplement to this comment:
http://www.earth-surf-dynam-discuss.net/esurf-2017-12/esurf-2017-12-RC2-supplement.pdf

---

## Referee Comment (RC3) · Anonymous Referee #3 · 13 May 2017

**General comments**

This manuscript presents a new approach for detecting and locating rockfalls using seismic signals, applied to a case study in the Swiss Alps. I find the manuscript well written, well organized and the results interesting. Validity and precision of the method have been carefully discussed, while I found the discussion about possible limitations a bit dry. I suggest to improve this part, especially given the fact that several manual adjustments and optimizations are needed in post-processing. Below are some minor comments about the main text and the supplementary materials.

**Specific comments on the main text.**

Title: I find the title too vague. The title should reflect that the manuscript is about one possible method for seismic rockfall monitoring, applied to a specific case study.

P. 3, Lines 19-20: this would be true if seismic waves were travelling in a homogeneous medium. When looking at high frequencies, like in this study, seismic waves are mostly sensitive to the crust and therefore their travel time is strongly affected by crustal heterogeneities and shallow slow velocity layers. Seismic tomographies of the Alps have shown crustal heterogeneities as large as 20

Figure 1: a large-scale map, showing the Lauterbrunner Vally on a larger context, would be informative.

Figure 2 and 4: the power spectral density is usually normalized to the frequency bin width and therefore the unit is $(m/s)^2/Hz$. Why this is not the case here?

P. 6, Line 23: how the length of the STA and LTA windows affects your results? How these two values have been chosen?

P.6, Line 28: the authors set the minimum cut-off frequency of the filter to 10 Hz, but in Table 1 they also showed that, after adjusting by hand the frequency range for location, 5 rockfalls over 10 are detected at minimum frequencies lower than 10 Hz. Please discuss this point.

P.7, Lines 13-14: criterion (iv) is basically the geometrical spreading, which is also characteristic of seismic waves generated by earthquakes.

P.7, Line 19: "windows of 1.4 and 1.1 s" are referred to what?

P.8, Line 4: "700 to 4000 m/s" is referred to which seismic phase?

P.10, Lines 2-3: how do you choose the STA/LTA threshold?

Table 2: the default frequency range varies from rockfall to rockfall. How it has been chosen?

P.14, Line 4: The signal-to-noise ratio is strongly related to the amplitude of ambient seismic noise, which may vary in time and space. I think it's difficult to find a correlation with the duration of the event (and in fact, the correlation coefficient r is pretty small).

Please discuss this point.

P. 14, Line 9: is 2700 m/s the velocity of S waves? Please, specify the seismic phase associated with the velocity here and everywhere in the paper.

P. 16, Line 9 and P.17, Line 28: please, define the threshold value using 3 digits or use the exponential notation.

P.17, Lines 14-16: it seems that, although the algorithm should work automatically, a lot of small manual adjustments are needed in order to get a precise location of the rockfalls. I encourage the authors to discuss more in detail this point, and not just in three lines. Manual adjustments imply a certain level of subjectivity and, in order to ensure reproducibility of the results, this limitation should be discussed carefully.

P. 18-19, section 6.3: a recent paper (Gualtieri and Ekstrom, 2017) discussed a similar rockfall behavior. Please discuss your findings in relation with this reference. In particular, they describe the first stage of a rockfall as related to the elastic rebound of the Earth following the mass detachment rather than to the opening and propagating of fracturing. Figure 2 also shows a strong signal at 9:03:48, potentially related to a fourth stage.

P.20, Line 17: sensu strictu should be sensu stricto.

**Specific comments on the supplementary materials.**

I have tested the code and I have two main remarks:

1) the .pdf with the detailed explanation of the code is very useful, but it would be also good to have the actual code (a file .R) in the folder.

2) The code worked as promised, except for the installation of the package "eseis". I had to download and install the package manually. I am working on a Mac OS v. 10.12.4 and I am using Rstudio v. 1.0.136.

**Suggested references.**

Diehl, T., Husen, S., Kissling, E., Deichmann, N. (2009). High-resolution 3-D P-wave model of the Alpine crust. Geophysical Journal International, 179(2), 1133-1147.

Gualtieri, L., and Ekström, G. (2017). Seismic Reconstruction of the 2012 Palisades Rockfall Using the Analytical Solution to Lamb's Problem. Bulletin of the Seismological Society of America, 107(1), 63-71.

---

## Referee Comment (RC4) · A. Helmstetter (Referee) · 31 May 2017

This manuscript applies two methods for studying rockfall activity in the Lauterbrunnen valley. Coupling seismic monitoring and terrestrial laser scanning (TLS) allows a good resolution in time and space, and allows the detection of very small rockfalls. TLS data is used to validate the seismic detection and location method.

While seismic monitoring and TLS have been frequently used, coupling both methods is innovative and interesting.

The authors obtain impressive results in terms of location accuracy and sensitivity. For these reasons this manuscript is very worth publishing in ESDD.

But some changes should be made to clarify a few points.

**-- Main points**

**- Detection and classification of seismic events**

p6-7. Events detected at different stations are considered to be the same event if the time delay between stations is less than 1.75 s corresponding to an S wave with a velocity of 2000 m/s.

I suggest increasing this value to about 10 s, because it is likely that some stations may detect the detachment phase, while other stations may only be triggered by the impact at the cliff base. Another possibility is to merge events at different stations if there is some overlap in time between the signals.

p13: Events longer than 20 s are removed because this is longer than the expected rockfall propagation.

But rockfalls frequently occur in sequences of events, so that this constrain may remove true rockfall events.

Here are a few ideas to distinguish automatically earthquakes and rockfalls :
- Did you use earthquake catalogs to remove earthquakes?
- The variability of amplitude among stations should be higher for rockfalls   ( or other nearby sources) than for earthquakes
- The time delay between stations should be smaller for earthquakes and other distant sources (deep source implying a higher apparent velocity).

**- Location.**

P7, l30. You should also cite here Lacroix and Helmstetter (2011) who used a very similar method to locate rockfalls (using the seismic waveforms rather than the signal envelope)

p8, l30. I do not understand "Locations with a likelihood quantile below 0.95 …"?
Do you mean that you select grid points with cross-correlation smaller than the 0.95 quantile of the distribution of the cross-correlation across the search area?
Or do you have a method to estimate the actual probability of a point to be the source location, e.g., as done by Lomax for the nonlinloc location algorithm?

I don't see the interest of adjusting the frequency range individually for each event. Of course, it makes the location error smaller.
By adjusting more parameters (time interval …) you could probably lower the location error event more …
But what do we learn from that?
Adjusting the velocity using all events makes sense, but adjusting one parameter for each event individually does not.
Even without optimizing the parameters based on known event location, the location accuracy is quite good considering the number of stations (between 4 and 6)!

Did you test your location method on synthetic signals?
For instance, you can take a real rockfall signal, and shift this signal in time by the difference in travel time to define the signal at the other stations, and add seismic noise.
This would provide an optimistic estimate of the location accuracy, because real signals are quite different from one station to another one. It can be useful to estimate the influence of errors on seismic wave velocity.

p17,l24 : The station spacing in your study is quite different from the study of Lacroix and Helmstetter (2011).
This study used antennas of 7-24 sensors, with distance between sensors inside an antenna of 20-50 m, and distances between antennas of several hundred meters.
Using shorter inter-sensor distance allows correlating the rockfall waveforms rather than their envelope and provides a better location accuracy.

- **Ambiguities in matching TLS and seismic events?**

p8l5. TLS locations are used to constrain the seismic wave velocity V used for locating the seismic events by minimizing the difference in location between TLS and seismic events.
Similarly, the frequency range used to filter the seismic signals is adjusted by minimizing the error with the TLS location.
But you already need an accurate location of the seismic events to associate seismic and TLS events!
Where did you start from? How did you deal with ambiguities? This part needs more explanations.

Maybe you could   select the rockfall seismic signal with the largest amplitude and assume it corresponds to the largest volume detected by TLS, and adjust the seismic wave velocity for this event?
Then run the location with this velocity for all events, associate TLS and seismic events, and only latter re-optimize V for all events?

**- Duration of events.**
In figure 7, the duration of signals does not seem to match the duration listed in Table 2. For instance, events #7 and #9 have duration >30 s when looking at the spectrograms, but the duration listed in Table 2 is much shorter.

Could you add symbols in each PSD plot showing the start and end of each event?

- Interpretation of seismic signals : impact or detachment?
p19. A figure showing a profile of the cliff at the location of the rockfall would be useful to interpret the rockfall signal.
Does the topography of the cliff supports the hypothesis of an impact 1.7 s after the initiation phase?

I think that the first low-frequency peak ("phase 2") is more likely the detachment phase (elastic rebound) than an impact.
Indeed, I have seen such a signal for many rockfalls that occurred under a   roof above an over-hanging cliff, with no possible impact before the cliff base, and with a time delay between the detachment phase and the impact at the base that is consistent with free fall.

You discuss only one event in section 6.3. What about the other 9 events?
Can you identify fracture, detachment, impact and/or propagation phases?
If you see both the detachment and impact phases, do you find a good agreement between the free fall height estimated from the seismic signal and from the source location?

**-- Details**

FIg 3: Add a scale bar and all plots

Fig 7 : For which station is the PSD computed?

Figure 5 : There are 5 solid lines corresponding to events with Pmax>0.94. But according to Fig 6 there should be 10 events with Pmax>=0.94?

Table 1 : can you add the number of available stations?

Table 2 : Could you also add magnitude (and/or amplitude range) for each rockfall?

---

## Author Response (AR1)

**Response to referee comments**

[Seismic monitoring of small alpine rockfalls  validity, precision and limitations]

July 13, 2017

We would like to thank the referee for the encouraging and helpful comments, all of them obviously devoted to improve the quality and impact of the manuscript.
* * *
**Referee 2.1**: *Title: When first reading the title of the paper I was expecting an analysis of the feasibility to study rockfalls with seismic methods in different contexts, or an enriched review of past studies on this subject. "Validity" and "limitations" of seismic rockfalls monitoring in general are not discussed in this paper and I find that what the authors propose is essentially an interesting case study. This should be explicit and clarified in the title. I would suggest for example: "Validity, precision and limitations of the seismic detection and location of small rockfalls in the Swiss Alps".*

**Reply**: We understand the arguments and changed the title (almost) as suggested.
* * *
**Referee 2.2**: *P2 L17: Please order the references chronologically. There is a wealth of studies on landslides seismic signals. If you decide to select some of them as examples, use "e.g." before citing them.*

**Reply**: All reference lists were checked for chronological order and corrected where necessary. The term "e.g." was inserted as suggested.
* * *
**Referee 2.3**: *P2 L18: While it would be an honor to share the name of David Hilbert, I am not. Here and everywhere else please correct the references to "Hibert et al." (no "L").*

**Reply**: Indeed, this was an unecessary bug that sneeked into the tex file. It has been corrected throughout.
* * *
**Referee 2.4**: *P2 L21: "cf" not necessary here or elsewhere.*

**Reply**: Terms have been removed where necessary/appropriate.
* * *
**Referee 2.5**: *P3 L14: Is this different from spectrograms?*

**Reply**: The term has been replaced by "spectrograms".
* * *
**Referee 2.6**: *P3 L19: Burtin et al. [2016] were not the first to show that seismic signals generated by rockfalls are dominated by surface waves. In the references here you can add Deparis et al., [2008], Dammeier et al., [2011], and Levy et al., [2015].*

**Reply**: References included as suggested.
* * *
**Referee 2.7**: *P3 L22: "Vital" seems a bit strong. Interesting? Significant? Crucial?*

**Reply**: Changed to "unique, important". We believe that this is the best phrase to describe the value of seismic data with respect to the level of detail they can provide in some cases, e.g., as shown in the example figure 2.
* * *
**Referee 2.8**: *Section 3: Is this section necessary? Could you move this into the introduction?*

**Reply**: We had thought about adding this chapter to the introduction during the writing process but then decided to keep it separate, mainly to adequately set the scope for the entire manuscript: i) "We know that seismic monitoring works for characterising rockfall, but there is a set of unknowns at the moment" and ii) "For those who are unfamiliar with the seismic approach, this is what the data looks like one can record and has to interpret". Furthermore, this section already presents results of this study, which makes it difficult to include it to the introduction. Thus, we prefer to keep the section in its current form, also backed up by no such impression by any of the three other referees.
* * *
**Referee 2.9**: *P3 L28: What is the "limit of detection"? Is it the targeted (or the possible?) resolution of the point clouds?*

**Reply**: This term is common jargon among the TLS community. We added a short definition in brackets for comprehension by a wider readership.
* * *
**Referee 2.10**: *P4 L3: Are the seismometers 3 components? Please add this information here. Figure 2 – caption: Do you know the volume of this*

*particular event? If so this information could be added in the caption. Are the signals filtered?*

**Reply**: Component information was added as suggested. Released rock volume, event ID and link to table 1 are provided in the caption, now. Filter window (1–90 Hz) is given in the caption, as well.
* * *
**Referee 2.11**: *P6 L17: The STA-LTA ratio picker was first proposed by Allen [1982].*

**Reply**: Reference was changed as suggested.
* * *
**Referee 2.12**: *P6 L18: Envelopes of seismic signals are commonly computed from the Hilbert transform of the signal. I think using the absolute amplitude is not a problem for detection, but for a localisation method based on the cross-correlation of envelopes,the Hilbert transform might yield better results. While I certainly do not think it is necessary to redo the current analysis with Hilbert envelopes, I would suggest testing this in future studies.*

**Reply**: Absolutely correct. The utilised algorithm used the Hilbert (with l this time) transform to calculate the envelope. The main idea in the original manuscript was to provide the unfamiliar reader with a short explanation of the term "envelope". However, obviously this plan was misleading. We removed the short and wrong definition, now.
* * *
**Referee 2.13**: *P6 L24: Can you indicate here what are the threshold values chosen?*

**Reply**: Since these values are part of the results, we provide here now the link to the adequate chapter (5.2), where these values are presented and justified.
* * *
**Referee 2.14**: *P7 L1 and 11: You choose here a velocity for S-waves but as stated before rockfalls seismic signals are dominated by surfaces waves (which are slower than body waves). How many events have you excluded based on this criterion?*

**Reply**: We have added further credit to earlier studies that point at the value of 2000 m/s for land slides and rock falls. Actually, after a test re-run of the approach on a short section of the data base with lower velocities,

all additionally included events were rain drop impacts (based on the short duration of the picks and relation to the meteorolgical data).
* * *
**Referee 2.15**: *P7 L10-11: This is a bit confusing. You first had an automated exclusion criterion based on the time delay between the onsets of the waves recorded at each station of the network and then you still check manually if this criterion is verified? What is the point of the first automated exclusion then? Maybe reorganize this paragraph and the one just before to improve clarity.*

**Reply**: Indeed, point i) and ii) are redundant. The initial idea was to have all decision/rejection criteria at one place as a summary. However, this was confusing. We removed these two points.
* * *
**Referee 2.16**: *P7 L13-14: Criterion iv): Does this imply that you know the location of the events before manually selecting the signals?*

**Reply**: The section was rephrased to be more general, it now simply expresses that the signals are expected to show a significant difference in their amplitudes due to the different source–receiver distances causing attenuation. If the source is inside the network, the differences between source and receiver for all possible station pairs is expected to be much higher than for a source location outside the network, especially if the source is away several times the network aperture, when only site amplification effects may modify the picture.
* * *
**Referee 2.17**: *P7 L17: References : "e.g." or add at least Surinach et al. [2005].*

**Reply**: Both suggestions were implemented.
* * *
**Referee 2.18**: *P7 L20: "Multiptaper" Typo?*

**Reply**: Yes, the typo has been corrected.
* * *
**Referee 2.19**: *P7 L21: References : "e.g.". Hibert et al. [2011] and Dammeier et al. [2011] seem more appropriate references here.*

**Reply**: Included/corrected as suggested.
* * *
**Referee 2.20**: *P7 L29-34: The approach proposed by Hibert et al. [2014 and not 2011] is designed to overcome all the issues regarding the specificity of rockfall seismic signals you enumerate before this sentence (emergent onset, waveform discrepancies, absence of seismic phases, high-frequency). Moreover if the dominant issue is the "differences between waveform properties at different stations" the cross-correlation approach would not work. In your case you can use a method based on the cross-correlation of the signal waveforms because the signals recorded at different stations are not too dissimilar. I suspect this is the case because the aperture of your network is not large (inter-station distances of 1 km). This is not the case at the Piton de la Fournaise volcano and is one of the difficulties that forced us to develop a new kurtosis-based first-arrival picker that is accurate enough to pick emergent signals. Please rewrite this paragraph by taking this remark into account.*

**Reply**: As suggested, the paragraph has been rewritten.
* * *
**Referee 2.21**: *P8 L19-21: Topography correction is necessary because rockfalls generate surface waves that propagate following the topography. Also the correct reference is Hibert et al. [2014] and not [2011] here.*

**Reply**: The text was changed as suggested and the reference was corrected.
* * *
**Referee 2.22**: *P8 L8-18: If you change the frequency range used to find the time lag that yields the best cross-correlations this should have an impact on the optimal velocity, and vice versa. Can you elaborate on the interdependence of the optimal frequency bands and the optimal velocities found? It can be interesting to add in Table 2 the velocity that gives the best location for these 10 rockfalls.*

**Reply**: As suggested, this section does now discuss the interconnectivity of wave velocity and frequency range used in the location routine. Since we kept the wave velocity constant for the different frequency bands there is limited value in adding it to the data table.
* * *
**Referee 2.23**: *P8 L21-22: Not all Earth surface processes generate seismic signal dominated by surface waves. I suggest to change "other Earth surface processes" into "other mass movement processes" or "gravitational processes". Also see comment on P3 L19 regarding the reference to Burtin et al., [2016].*

**Reply**: Changed as suggested.

**Referee 2.24**: *P8 L22-25: It is not clear how you performed this correction. What is: "that part where direct distance is above the actual surface elevation"? Does this mean that you corrected the direct straight line distance from pixel to pixel by the slope angle? Did you compute profiles for each pixel-station pair from the intersection of the straight line between those two points with the grid points of your DEM? Integrating the topography in propagation maps is not a trivial task but as you mentioned is critical to have accurate locations of rockfalls. This should be a bit more detailed, especially if the main focus of this paper is the capability to locate rockfalls from the seismic signal they generate.*

**Reply**: We added further explaining sentences. The approach is a direct translation of the Matlab based technique discussed by Burtin et al. (2014, ESurf) to the language R and is part of the freely available package eseis.

**Referee 2.25**: *P8 L30: Please define what is the "likelihood quantile" before.*

**Reply**: This value is now defined at this position and used throughout the text. $P$ is the location cross-correlation value of a given pixel and the 0.95 quantile is the threshold value arising from all $P$ values of the location grid used to, e.g., clip the location polygons.

**Referee 2.26**: *P9 L26: What caused this tilting? Do you know when it started? What is the influence of this tilting on the seismic signals recorded before dismantling those stations?*

**Reply**: The (most likely) cause of the tilting is now mentioned in the text. It is hard to say when this started because the TC120s sensors can compensate tilting up to about 10 degrees by using additional battery power but suddenly fail to record further data once beyond this tilting angle. Anyhow, we think this technical detail about the utilised sensor is of limited use for the reader and prefer not to infuse it into the text.

**Referee 2.27**: *P11 L16-18: I am a bit sceptical regarding the "rain drop sources" because you have buried your stations at 30-40 cm depth. This should prevent any direct contact between the seismometers and rain drops and I think rain drops are too weak seismic source to generate signals that would not be attenuated in the first few centimeters of propagation. Other common sources that can generate impulsive signals with energy in high-frequency bands are thunder, numerical glitches or close footsteps (animal or human). You based your attribution of those signal to rain drops from*

*the observation that "it only occurs in the records when it was raining in the Lauterbrunnen Valley during deployment and maintenance of stations". So you observe these noise signals on the days you were on the sites. There is a possibility that these signals are your footsteps, but without clear evidences we do not know. So did you observed those signals on days where you were not on site? Can you provide other arguments to attribute those signals to rain drops? For example, did you observe that those signals appeared and disappeared gradually over a period of times of several tens of minutes (or few hours), mimicking the passing of scattered showers? If so could you show this on a figure to definitely convince your readers that those signals are indeed generated by rain drops? If not I would suggest to rename this class of source to "impulsive noise".*

**Reply**: We added a further figure showing the co-occurrence of the seismic signal pulses and a rain data record. The data is also interpreted in the text and we argue for the rain cause with respect to passing animals or humans on the base of the irrgularity of the signals.
* * *
**Referee 2.28**: *P14 L3-4: What is "n"? What is "r"?*

**Reply**: "$n = 8$" has been replaced by "eight cases" and "r" has been removed completely, see comments of referee three.
* * *
**Referee 2.29**: *P14 L4: If you want to provide to the readers an analysis based on the SNR you need to indicate how you have computed this quantity before.*

**Reply**: The term SNR is now defined where it is used for the first time (chapter 4.3).
* * *
**Referee 2.30**: *P17 L13-14: What are those relationships? Do you refer to the studies of Hibert et al. [2014], Manconi et al. [2016]? Dammeier et al. [2016]?*

**Reply**: The statement is now supported by some of the suggested references.
* * *
**Referee 2.31**: *P17 L20-21: Levy et al. [2015] used a different approach (first-arrival picking and not cross-correlation back propagation as in the present study) and had a network with a much larger aperture (inter-stations distances of few kilometers).*

**Reply**: Corrected as suggested.

**Referee 2.32**: *P18 L7-9: I agree with the assumption that a rockfall with a higher volume should generate a higher-amplitude seismic signal if the travelled path and the fall height are the same. However you say latter that in your case there are no correlation between seismic energy/amplitude and the volumes of the events.*

**Reply**: We rewrote the statement further down to say explicitly that it refers to relationships based on volume, only. In the Lauterbrunnen Valley case we would have to include many more parameters than just rock volume, as explained in the rest of this paragraph. So in summary, both parts are true: in the case where only the rock volume is different while all other parameters (e.g., height, fragmentation, debris entrainment and impact location) are identical, the seismic signal of a larger rock mass will undoubtly be larger. But when all the other parameters can vary, as well, this energy-volume relationship will fade.
* * *
**Referee 2.33**: *P18 L12-13: The volume of the rockfalls in the study by Hibert et al. [2011] had volumes as low as few cubic meters.*

**Reply**: The sentence was rephrased and more appropriate references were used, now.
* * *
**Referee 2.34**: *P18 L18-19: Indeed, you are working with complicated events and I acknowledge that extracting quantitative laws might be difficult in this case. However, as shown by the example discussed in section 6.3, you are able to identify the different stages of the rockfall propagation. Is this true for the 10 rockfalls in your database? If so, you have every information you need (location of the events, volumes/masses, average velocity of the medium) to go further in your analysis. For example, what are the relationships between the first impulsive arrival amplitude (corrected from propagation effect) (phase 1) and the volume? The relationships between the seismic energy and the potential energy lost at the first impact with the topography (phase 2) ? The same relationships during the propagation phase on the talus (phase 3)? Those are fundamental issues that you might be able to contribute to answer with your dataset. Even if no relationships are found, this would still be very interesting as it will nourish discussion on the validity for small rockfalls of the relationships found by others [e.g. Deparis et al., 2008; Vilajosana et al., 2008; Hibert et al., 2011; Dammeier et al., 2011; Yamada et al., 2012; Ekström and Stark, 2013; Farin et al., 2015; Levy et al., 2015; Hibert et al., E-Surf in press]. I understand that this might be out of the scope of this study, but I think that adding this deeper analysis will significantly improve the impact and the reach of your paper.*

**Reply**: Indeed, for some of the events there is a comparably favourable situation as for the event shown in figure 2, but this would reduce the number of suitable cases to about four. We believe this is not a sufficient amount of data to hypothesise about quantitative laws. As suggested by the referee, the topic is out of the manuscript scope and extending the discussion to this theme would inevitably require a significant redesigning process of the entire mansucript, a point we consider not balanced by the number of suitable events that can be used to support claims in the light of this goal. We have however opened the door for the reader to think about this possibility at the end of the paragraph (last two sentences).
* * *
**Referee 2.35**: *P18 L31: references : add "e.g." and/or other references, for example : Helmstetter and Garambois [2010], Yamada et al., [2012], Zimmer and Sitar., [2015], Hibert et al. [2017].*

**Reply**: Both suggestions were included.
* * *
**Referee 2.36**: *P19 L3-5: While it seems reasonable to think that large mass detachments are preceded by cracking and fracture opening that generates an increasing rate of micro-earthquakes, this is more debatable for very small rockfalls such as the ones in this study. Another assumption to explain this first impulsive signal is that it is generated by the rebound of the Earth in the departure zone due to the detachment of the mass. This was observed at Piton de la Fournaise volcano [Hibert et al., 2011]. I think both assumptions should be mentioned here.*

**Reply**: Implemented as suggested.
* * *
**Referee 2.37**: *P19 16-17: If larger particles have higher momentum they will reach the bottom of the slope more rapidly than small particles. In fact this is what is observed in many cases on video recordings of events: large blocks preceding the flow of small granular materials. The loss of high-frequency at the end of seismic signals generated by gravitational instabilities is complex and still not yet fully understood. Analytical models [e.g. Okal, 1990; Farin et al., 2015] suggest that events with larger volume will indeed generate signal with a lower corner frequency, but the overall amplitude of the signal across the whole frequency range will be higher. To this adds the fact that high frequencies generated by small particles are more attenuated. The combination of those two processes suggests that the loss of high-frequency at the end of those seismic signals is due to the early immobilization of the largest particles, not the smallest. This is highly speculative, and any*

*interpretation of this frequency shift has to be done carefully and supported by data. If you want to comment on this, please add references.*

**Reply**: As this part of the anatomy section is not a central part of the scope of the manuscript we followed the referee suggestion. We removed the speculative part and provide a reference to support the first part.

**Response to referee comments**

[Seismic monitoring of small alpine rockfalls  validity, precision and limitations]
July 13, 2017

We would like to thank the referee for the encouraging and helpful comments, all of them obviously devoted to improve the quality and impact of the manuscript.
* * *
**Referee 3.1**: *This manuscript presents a new approach for detecting and locating rockfalls using seismic signals, applied to a case study in the Swiss Alps. I find the manuscript well written, well organized and the results interesting. Validity and precision of the method have been carefully discussed, while I found the discussion about possible limitations a bit dry. I suggest to improve this part, especially given the fact that several manual adjustments and optimizations are needed in post-processing. Below are some minor comments about the main text and the supplementary materials. Specific comments on the main text.*

**Reply**: We reorganised the discussion and especially the conclusion chapter to highlight the limitation of the seismic method. Also, the need for manual supervision has been added to the abstract, introduction and discussion chapter 6.1 (also see referee comment 17).
* * *
**Referee 3.2**: *Title: I find the title too vague. The title should reflect that the manuscript is about one possible method for seismic rockfall monitoring, applied to a specific case study.*

**Reply**: As suggested, also by referee 2, the title has been focused with respect to location and event size.
* * *
**Referee 3.3**: *P. 3, Lines 19-20: this would be true if seismic waves were travelling in a homogeneous medium. When looking at high frequencies, like in this study, seismic waves are mostly sensitive to the crust and therefore their travel time is strongly affected by crustal heterogeneities and shallow slow velocity layers. Seismic tomographies of the Alps have shown crustal heterogeneities as large as 20.*

**Reply**: The additional information is now included, making lcear that the homogeneous medium is an idealised case and that under natural conditions

there can be significant alterations.
* * *
**Referee 3.4**: *Figure 1: a large-scale map, showing the Lauterbrunnen Valley on a larger context, would be informative.*

**Reply**: Overview map of Switzerland and the location of the study area therein has been added.
* * *
**Referee 3.5**: *Figure 2 and 4: the power spectral density is usually normalized to the frequency bin width and therefore the unit is (m/s)/Hz. Why this is not the case here?*

**Reply**: The missing legend item has been added.
* * *
**Referee 3.6**: *P. 6, Line 23: how the length of the STA and LTA windows affects your results? How these two values have been chosen?*

**Reply**: Indeed, the window length obviously affect the number and timing of initially picked events. We added an explaining sentence for clarification. Our reasoning for chosing these values is and was based on the referenced study of Burtin et al. (2014), as stated in the manuscript. It is hard to inspect how different values of STA and LTA window lengths would affect the results because the picking is just the start of a long chain of further steps to remove spurious events. We think adding information of other window lengths would add little further information to trace the overall contribution to the final number of events.
* * *
**Referee 3.7**: *P. 6, Line 28: the authors set the minimum cut-off frequency of the filter to 10 Hz, but in Table 1 they also showed that, after adjusting by hand the frequency range for location, 5 rockfalls over 10 are detected at minimum frequencies lower than 10 Hz. Please discuss this point.*

**Reply**: Indeed, the filter frequency window for location is different than for picking. For picking, the main goal is to have it matching most of the envisioned events and not to already focus on the actual frequency content of the individual rockfalls. We believe that this point is clear by referring to the four studies that noted the typical frequency content of rockfalls. The referee is absolutely correct that the filter frequencies to optimise the location estimate are lower than 10 Hz for many events. We comment on this point now in section 6.1.
* * *
**Referee 3.8**: *P. 7, Lines 13-14: criterion (iv) is basically the geometrical spreading, which is also characteristic of seismic waves generated by earthquakes.*

**Reply**: The criterion (now cirterion ii) has been clarified to point at the difference of a source inside the network versus a source far away from the network.
* * *
**Referee 3.9**: *P. 7, Line 19: "windows of 1.4 and 1.1 s" are referred to what?*

**Reply**: The term "moving time windows of 1.4 and 1.1 s to generate the spectra" has been added for clarification.
* * *
**Referee 3.10**: *P. 8, Line 4: "700 to 4000 m/s" is referred to which seismic phase?*

**Reply**: The term has been replaced by "apparent velocities" and essential references to support it, see comments of referee 2.
* * *
**Referee 3.11**: *P. 10, Lines 2-3: how do you choose the STA/LTA threshold?*

**Reply**: The values were not chosen from best guesses but are based on the measured waveforms of the control events. The link to this sentence before the statement is now strengthened by linking these two sentences.
* * *
**Referee 3.12**: *Table 2: the default frequency range varies from rockfall to rockfall. How it has been chosen?*

**Reply**: Please see the justifications in section 4.4, third paragraph.
* * *
**Referee 3.13**: *P. 14, Line 4: The signal-to-noise ratio is strongly related to the amplitude of ambient seismic noise, which may vary in time and space. I think it's difficult to find a correlation with the duration of the event (and in fact, the correlation coefficient r is pretty small). Please discuss this point.*

**Reply**: We removed the discussion of the SNR relationships, since – as the referee points out – it is difficult to find these relationships. Furthermore, the information is far from being essential for the scope of the article.
* * *
**Referee 3.14**: *P. 14, Line 9: is 2700 m/s the velocity of S waves? Please, specify the seismic phase associated with the velocity here and everywhere in the paper.*

**Reply**: As also pointed out by reviewer 2, we clarified the term to "apparent velocity" throughout the manuscript and give adequate references to underline that for such signals it is often not possible to identify the different phases.
* * *
**Referee 3.15**: *P. 16, Line 9 and P.17, Line 28: please, define the threshold value using 3 digits or use the exponential notation.*

**Reply**: As suggested, we rounded to three digits.
* * *
**Referee 3.16**: *P. 17, Lines 14-16: it seems that, although the algorithm should work automatically, a lot of small manual adjustments are needed in order to get a precise location of the rockfalls. I encourage the authors to discuss more in detail this point, and not just in three lines. Manual adjustments imply a certain level of subjectivity and, in order to ensure reproducibility of the results, this limitation should be discussed carefully.*

**Reply**: In fact the described workflow is not intended to result in a recipe for automatic rockfall detection and location, especially not for such small events as faced in this study. We added a clarifying sentence about this scope now at the beginning of section 6.1 and also at the end of the introduction and the abstract.
* * *
**Referee 3.17**: *P. 18-19, section 6.3: a recent paper (Gualtieri and Ekstrom, 2017) discussed a similar rockfall behavior. Please discuss your findings in relation with this reference. In particular, they describe the first stage of a rockfall as related to the elastic rebound of the Earth following the mass detachment rather than to the opening and propagating of fracturing. Figure 2 also shows a strong signal at 9:03:48, potentially related to a fourth stage.*

**Reply**: The article by Gualtieri and Ekstrom (2017) focuses on an event about $10^4$ m$^3$, which is very different from the rockfalls our study focuses on, mainly below $10^0$ m$^3$. The potential source of the signal due to the elastic rebound of the cliff after detachment is now discussed in chapter 6.3, cf. comments of referee 2.
* * *
**Referee 3.18**: *P. 20, Line 17: sensu strictu should be sensu stricto.*

**Reply**: The term has been corrected.
* * *
**Referee 3.19**: *I have tested the code and I have two main remarks: 1) the pdf with the detailed explanation of the code is very useful, but it would be also good to have the actual code (a file .R) in the folder. 2) The code worked as promised, except for the installation of the package eseis. I had to download and install the package manually. I am working on a Mac OS v. 10.12.4 and I am using Rstudio v. 1.0.136.*

**Reply**: We are thankful for the invested time to reproduce the results of the study using the same software we used. Initially we considered adding also the set of R scripts to the supplementary materials. However, each fo the about 8 scripts contains hundreds of code lines and would require significant manual adjustments of paths and are actually optimised to automatically create the figures of the manuscript. Thus, they are not optimised for comprehension but for performance.

The installation issue is know to the main author, and related to both the Mac OSX and the current way to host the package on the website. It is intended to release the package on the Comprehensive R Archive Network (CRAN), which will fix the problem. Meanwhile, additional information about installing the package manually is provided on the website of the first author (`http://www.micha-dietze.de/pages/eseis.html`).

**Response to referee comments**

[Seismic monitoring of small alpine rockfalls  validity, precision and limitations]

July 13, 2017

We would like to thank the referee for the encouraging and helpful comments, all of them obviously devoted to improve the quality and impact of the manuscript.

**Referee 4.1**: *This manuscript applies two methods for studying rockfall activity in the Lauterbrunnen valley. Coupling seismic monitoring and terrestrial laser scanning (TLS) allows a good resolution in time and space, and allows the detection of very small rockfalls. TLS data is used to validate the seismic detection and location method. While seismic monitoring and TLS have been frequently used, coupling both methods is innovative and interesting. The authors obtain impressive results in terms of location accuracy and sensitivity. For these reasons this manuscript is very worth publishing in ESDD. But some changes should be made to clarify a few points.*

**Reply**: We are thankful for the encouraging feedback and refer to the changes as suggested by the points below.

**Referee 4.2**: *p6-7. Events detected at different stations are considered to be the same event if the time delay between stations is less than 1.75 s corresponding to an S wave with a velocity of 2000 m/s. I suggest increasing this value to about 10 s, because it is likely that some stations may detect the detachment phase, while other stations may only be triggered by the impact at the cliff base. Another possibility is to merge events at different stations if there is some overlap in time between the signals.*

**Reply**: In an earlier stage of the project we pursued this concept. However, location of the rockfall events in this study is only possible when the same seismic source (e.g., detachment process or impact) is recorded by all (at least four) stations. Allowing for larger time windows would indeed cause triggering of different event phases by different stations and thus, at best, a smearing of the location estimate. Thus, we need to keep this narrow time window. We explain this necessity now in the STA/LTA paragraph of the revised manuscript.

**Referee 4.3**: *p13: Events longer than 20 s are removed because this is longer than the expected rockfall propagation. But rockfalls frequently occur in sequences of events, so that this constrain may remove true rockfall events. Here are a few ideas to distinguish automatically earthquakes and rockfalls: Did you use earthquake catalogs to remove earthquakes? The variability of amplitude among stations should be higher for rockfalls (or other nearby sources) than for earthquakes. The time delay between stations should be smaller for earthquakes and other distant sources (deep source implying a higher apparent velocity).*

**Reply**: Correct, rockfalls – also some of the events described in this manuscript – consist of sequences of activity, including talus slope mobilisation (e.g., event 8). However, the constraint of 20 s is only used for the STA/LTA picker phase. Sequences of releases would result in several subsequent but short STA/LTA picks, as shown in figure 4b of the manuscript. The earthquake catalogues certainly contain the large events during the instrumented period. But smaller, still not rockfall-related events might not be contained in them. We initially tested the variance of signal amplitudes among the stations as a discriminator for earthquakes versus rockfalls but found that the power of this criteria is not high. Indeed, the time delay between stations is usually smaller for earth quakes than for rockfalls. But the duration criterion performed very well in our case, as shown by the average values in the discussed in the text. These points are now mentioned or discussed in the manuscript (chapter 4.3).
* * *
**Referee 4.4**: *P7, l30. You should also cite here Lacroix and Helmstetter (2011) who used a very similar method to locate rockfalls (using the seismic waveforms rather than the signal envelope)*

**Reply**: Included as suggested.
* * *
**Referee 4.5**: *p8, l30. I do not understand "Locations with a likelihood quantile below 0.95 ..."? Do you mean that you select grid points with cross-correlation smaller than the 0.95 quantile of the distribution of the cross-correlation across the search area? Or do you have a method to estimate the actual probability of a point to be the source location, e.g., as done by Lomax for the nonlinloc location algorithm?*

**Reply**: We mean the quantile concept. This is now expressed clearly in the manuscript, chapter 4.4, paragraph 5. As suggested by referee 2, the entire section has been revised to clarify this and other points related to better explain the location approach.

**Referee 4.6**: *I don't see the interest of adjusting the frequency range individually for each event. Of course, it makes the location error smaller. By adjusting more parameters (time interval ...) you could probably lower the location error event more... But what do we learn from that? Adjusting the velocity using all events makes sense, but adjusting one parameter for each event individually does not. Even without optimizing the parameters based on known event location, the location accuracy is quite good considering the number of stations (between 4 and 6)!*

**Reply**: The main point we want to work out with this exercise is to explore the highest possible location precision available with the data, methodology and landscape setting in this study. Tweaking any other parameters did actually not help improving the location estimate. Obviously, there will never be the chance to go beyond the "unimproved" location frequency approach. We mention this clarification now in the location chapter.
* * *
**Referee 4.7**: *Did you test your location method on synthetic signals? For instance, you can take a real rockfall signal, and shift this signal in time by the difference in travel time to define the signal at the other stations, and add seismic noise. This would provide an optimistic estimate of the location accuracy, because real signals are quite different from one station to another one. It can be useful to estimate the influence of errors on seismic wave velocity.*

**Reply**: Although such a test with synthetic data would shed more light onto the algorithm performance and capabilities, this manuscript rather follows the natural scale experiment setup. The location approach itself has been discussed in previous studies (cf. Burtin et al., 2013, 2016) and the R-version is a mere translation of it from the Matlab script by Arnaud Burtin and has been validated against this script before releasing the package. See also comment by referee 2.
* * *
**Referee 4.8**: *p17,l24 : The station spacing in your study is quite different from the study of Lacroix and Helmstetter (2011). This study used antennas of 7–24 sensors, with distance between sensors inside an antenna of 20–50 m, and distances between antennas of several hundred meters. Using shorter inter-sensor distance allows correlating the rockfall waveforms rather than their envelope and provides a better location accuracy.*

**Reply**: The unsuitable reference has been removed to correct the context of the sentence.

**Referee 4.9**: *p8l5. TLS locations are used to constrain the seismic wave velocity V used for locating the seismic events by minimizing the difference in location between TLS and seismic events. Similarly, the frequency range used to filter the seismic signals is adjusted by minimizing the error with the TLS location. But you already need an accurate location of the seismic events to associate seismic and TLS events! Where did you start from? How did you deal with ambiguities? This part needs more explanations. Maybe you could select the rockfall seismic signal with the largest amplitude and assume it corresponds to the largest volume detected by TLS, and adjust the seismic wave velocity for this event? Then run the location with this velocity for all events, associate TLS and seismic events, and only latter re-optimize V for all events?*

**Reply**: Indeed, the section was confusing. One misleading part arose from the optimised frequency section. This has been rewritten, see above. For the rest, in principal we treated our data and approach as if there were no independent TLS-based location constraints. Thus, the second paragraph of chapter 4.4 has also been rewritten to clarify.
* * *
**Referee 4.10**: *In figure 7, the duration of signals does not seem to match the duration listed in Table 2. For instance, events 7 and 9 have duration > 30 s when looking at the spectrograms, but the duration listed in Table 2 is much shorter. Could you add symbols in each PSD plot showing the start and end of each event?*

**Reply**: The issue has been clarified by explicitly mentioning when a duration was based on the STA-LTA-ratio method (cf. end of section 5.2) versus manually inspecting the waveforms (cf. caption of table 2). In the caption of figure 7 we have now added information about the time axis, i.e., that event start is indicated by the zero tick and duration can be found in table 2, since the end is mainly after a few seconds, which would be tricky to visualise in this figure. The definition of duration does not include subsequent slope activity (e.g., event 8).
* * *
**Referee 4.11**: *Interpretation of seismic signals: impact or detachment? p19. A figure showing a profile of the cliff at the location of the rockfall would be useful to interpret the rockfall signal. Does the topography of the cliff supports the hypothesis of an impact 1.7 s after the initiation phase? I think that the first low-frequency peak ("phase 2") is more likely the detachment phase (elastic rebound) than an impact. Indeed, I have seen such a signal for many rockfalls that occurred under a roof above an over-hanging cliff, with no possible impact before the cliff base, and with a time delay between*

*the detachment phase and the impact at the base that is consistent with free fall.*

**Reply**: The interpretation has been widened to include the possibility of rock detachment and cliff rebound (see also comments by referee 2). The cliff geometry with a few small ledges along the about 88 degree steep topography has been added to the study area description. The main point that would argue against phase 2 being the rebound is that the rock mass, as it reaches the cliff base, does not result in a single strong signal but rather a emergent wave form, which we tend to interpret as a shower of already fragmented rocks. This fragmentation must take place somewhere, and most probably this is phase 2 when the detached rock mass hits the cliff higher up.
* * *
**Referee 4.12**: *You discuss only one event in section 6.3. What about the other 9 events?*

**Reply**: Indeed, we could discuss other than the one example event. We chose to spotlight only this rockfall because the scope of the manuscript is on comparing the TLS data with the seismic detection and location results to pursue the goals mentioned in the title. A more thorough and rich discussion of seismic insight to rockfall can be found in another manuscript by Dietze et al., also submitted to ESurfD: Dietze, M., Turowski, J. M., Cook, K. L., and Hovius, N.: Spatiotemporal patterns and triggers of seismically detected rockfalls, Earth Surf. Dynam. Discuss., `https://doi.org/10.5194/esurf-2017-20`, in review, 2017. We think that including further rockfalls into this discussion section here would blur the focus or make it difficult to define which ones to include and which ones not. We provide the full seismic data to invite readers to reproduce and explore the properties of other events.
* * *
**Referee 4.13**: *Can you identify fracture, detachment, impact and/or propagation phases? If you see both the detachment and impact phases, do you find a good agreement between the free fall height estimated from the seismic signal and from the source location?*

**Reply**: Discriminating fracture from detachment is hardly possible with the data of this study, as discussed two points above. Based on the inferred free fall phase explained in figure 2 we indeed find a reasonable agreement of TLS-based detachment height and the cliff base/talus slope, a point we discuss in the second paragraph of chapter 6.3. However, the other manuscript (reference see above) gives a much deeper and more appropriate insight to this topic, based on a larger data set.

**Referee 4.14**: *Fig 3: Add a scale bar and all plots*

**Reply**: The images in a) and b) are perspective views, not orthorectivied imagery, which makes it difficult to add globally valid scale bars. We mention now in the figure caption the approximate extent of the cliff stretch and link to figure 1a) where the station distances are given. For c) the pixel sizes are not equal but modified to scale the detachment area for the plot.
* * *
**Referee 4.15**: *Fig 7 : For which station is the PSD computed?*

**Reply**: Information has been added to the figure caption.
* * *
**Referee 4.16**: *Figure 5 : There are 5 solid lines corresponding to events with $P_{max} > 0.94$. But according to Fig 6 there should be 10 events with $P_{max} >= 0.94$?*

**Reply**: The usage of the 0.95 quantile threshold and $P_{max}$ was confusing and has been clarified throughout the manuscript, see also comments by other reviewers. With respect to figure 5, there are only five lines because these are the only ones that reached an $R^2$ for the location estimate above 0.94. This value may not be confused with the quantile threshold used to clip the location estimate polygons, such as the 0.95 quantile or the 0.973 quantile. In the figure, 0.94 was used because the other lines reached significantly lower ($< 0.8$) values and were not used for the velocity estimate approach. In summary, $P_{max}$, $\Delta P_{max}$, $R^2$ and the 0.95 (or 0.973) quantile are now explicitly defined in chapter 4.4 and consistently used throughout the manuscript.
* * *
**Referee 4.17**: *Table 1 : can you add the number of available stations?*

**Reply**: Basically, the number for all events was four. The overlapping period, when five stations were operating yielded no rockfall event. This number of four stations is now explicitly mentioned in the beginning of chapter 5.2.
* * *
**Referee 4.18**: *Table 2 : Could you also add magnitude (and/or amplitude range) for each rockfall?*

**Reply**: Table column added as suggested.

[revised manuscript text omitted]

---

## Author Response (AR2)

**Response to associate editor comments**

[Seismic monitoring of small alpine rockfalls – validity, precision and limitations]
September 7, 2017
* * *
**AE 1**: *This is an interesting paper on the use of seismic methods for the detection of small-magnitude rockfall validated using multi-temporal TLS surveys. This manuscript has already undergone a great improvement after implementing most of the recommendations expressed by the reviewers. However, there are several points that still need to be addressed mainly concerned with the structure and writing of the manuscript. Overall, this can be considered as a moderate revision. Please find the details in the attached file with my comments over the revised manuscript. The main aspects that need improvement are the following:*

**Reply**: The suggestions in the annotated pdf file were revised to the extent the consulted native speakers confirmed their correctness or when technical modifications were sound.
* * *
**AE 2**: *1. In some places the structure is confusing, especially regarding section 3, 4.3 and the conclusions.*

**Reply**: Section 4.3 was expanded by providing a flow chart of data handling. Conclusions were restructured according to suggestions. For section 3 please see below.
* * *
**AE 3**: *a) Section 3 should be moved to the introduction as one of the reviewers already suggested and Fig 2 should be included on a aproppriate section.*

**Reply**: For the suggestion to move section 3 to the introduction, please see our argumentation regarding comment 8 of referee 2. Moving this section would inappropriately inflate the introduction and spoil its main role: justifying the study. Likewise, presenting figure 2 as part ofthe results the scope of the article would blur. In this article we do not intend to discuss event evolution analysis, mainly because we want to show the overall validity of the approach and ten events are not a sufficient size to draw robust conclusions. Thus, we decided early in the writing process of the manuscript to present one explanatory example on what seismic monitoring delivers and how this can be interpreted.

**AE 4**: *b) Section 4.3 could be improved by explaining more carefully the different stages, parameters and threshold values of the manual and automated detection procedure. This is critical since many of the questions of the reviewers ask for clarification on these aspects and it is a central aspect of this contribution. I strongly recommend depicting a diagram in which this quite complex methodology is clearly explained. Since currently there are 8 figures, including one more is reasonable. The authors have incorporated the different suggestions of the reviewers in the text but a proper graphical view of the multiple stages is really necessary.*

**Reply**: Figure is added. Most of the referee comments concerned justification of the parameters and clarifying the basis for the set values.

**AE 5**: *c) Regarding the conclusions, a clearer division in advantages, limits and future lines could be performed.*

**Reply**: Done as suggested.

**AE 6**: *2. The writing must be improved in many places. There are missing colons, hyphens and articles throughout the manuscript and many expressions do not sound correct, even for a non-native english speaker like myself. It is clear that the original or revised version of the manuscript have not been carefully revised by a native english speaker. This correction should be carried out to achieve a suitable version for publication.*

**Reply**: Two native speakers were consulted and changes were implemented when appropriate.